# FIGHTER: UNVEILING THE GRAPH CONVOLUTIONAL NATURE OF TRANSFORMERS IN TIME SERIES MODELING

## ABSTRACT

Transformers have achieved remarkable success in time series modeling, yet their internal mechanisms remain opaque. This work demystifies the Transformer encoder by establishing its fundamental equivalence to a Graph Convolutional Network (GCN). We show that in the forward pass, the attention distribution matrix serves as a dynamic adjacency matrix, and its composition with subsequent transformations performs computations analogous to graph convolution. Moreover, we demonstrate that in the backward pass, the update dynamics of value and feed-forward projections mirror those of GCN parameters. Building on this unified theoretical reinterpretation, we propose **Fighter** (Flexible Graph Convolutional Transformer), a streamlined architecture that removes redundant linear projections and incorporates multi-hop graph aggregation. This perspective yields an explicit and interpretable representation of temporal dependencies across different scales, naturally expressed as graph edges. Experiments on standard forecasting benchmarks confirm that Fighter achieves competitive performance while providing clearer mechanistic interpretability of its predictions.

## 1 INTRODUCTION

Time series data is prevalent in the current data-centric world. Effective modeling of time series is essential for informed decision-making and strategic planning across various domains, including finance (Tsay, 2005), transportation (Vlahogianni & Karlaftis, 2013), and energy management (Wang et al., 2007). The success of Transformer models (Vaswani et al., 2017) in natural language processing has quickly drawn the attention of researchers working on time series (Wen et al., 2023). Thanks to their ability to capture long-range dependencies, Transformer-based models have emerged as promising solutions for overcoming common challenges in time series modeling.

However, the internal workings of Transformers remain poorly understood (Rudin, 2019; Zeng et al., 2023) in time series, highlighting the need for a deeper investigation:

- *How does the Transformer process input time series data*? Specifically, what role do attention mechanisms and feed-forward networks (FFNs) play?

- *What are the update dynamics of Transformers*? In particular, how are the parameters in the attention mechanisms and FFNs updated?

Understanding these questions is crucial for the reliable application of Transformers in time series data, such as finance (Masini et al., 2023) and healthcare (Wickstrøm et al., 2020; Morid et al., 2023), and will offer insights into the design of more effective model architectures.

To this end, this work presents a unified theoretical reinterpretation that establishes a fundamental equivalence between the Transformer encoder and Graph Convolutional Networks (GCNs) (Kipf & Welling, 2017). GCNs are powerful architectures for modeling graphs, defined by a static adjacency matrix and feature matrix that represent fixed edge connections and node attributes, respectively (Hamilton et al., 2017; Chami et al., 2022). A GCN operates similarly to a multilayer perceptron (MLP), with its key advancement lying in aggregating features across multiple hops based on the adjacency matrix at the start of each layer (Wu et al., 2019; Zhang et al., 2025). We demonstrate that, in the forward pass, the Transformer attention distribution matrix (*i.e.*, $\text{softmax}(\boldsymbol{Q}\boldsymbol{K}^\top/\sqrt{d} + \boldsymbol{M})$,

where $Q$, $K$, and $M$ denote query, key, and mask matrices) serves as a dynamic counterpart to the adjacency matrix for feature aggregation. Unlike the static adjacency matrix used in GCNs, which corresponds to fixed graph connections, this distribution matrix is learnable via the parameters of the query and key matrices. The subsequent linear projections in the value matrix and feed-forward network (FFN)[1] perform computations that resemble MLP-based feature extraction in GCNs. During training, we show that the updates to the value matrix and FFN parameters mirror those in GCNs, focusing on feature extraction. Meanwhile, the parameters of $Q$ and $K$ facilitate an adaptive update, which is responsible for a dynamic and learnable adjacency matrix, in contrast to the static connections involved in GCNs.

Building on this unified perspective, we propose **Fighter** (Flexible Graph Convolutional Transformer), a streamlined architecture that removes redundant linear projections and integrates multi-hop graph aggregation for an explicit and interpretable representation of temporal dependencies across various scales. Lastly, experiments on standard forecasting benchmarks confirm that Fighter achieves competitive performance while offering clearer mechanistic interpretability of its predictions. Our key contributions are listed as follows:

- We establish a fundamental theoretical equivalence between the Transformer encoder and Graph Convolutional Networks (GCNs), revealing that the Transformer attention distribution matrix acts as a dynamic adjacency matrix for feature aggregation, while parameter updates to the value matrix and FFN during training mirror GCN dynamics. This unified framework demystifies the internal workings of Transformer encoders in time series modeling, bridging sequence and graph paradigms.

- We propose Fighter (Flexible Graph Convolutional Transformer), a novel architecture that leverages this equivalence by eliminating redundant linear projections and integrating multi-hop graph aggregation, enabling explicit and interpretable representations of multi-scale temporal dependencies as graph edges.

- We empirically evaluate Fighter on standard time series forecasting benchmarks, demonstrating its ability to effectively model temporal dependencies while providing enhanced interpretability through graph-based insights, as evidenced by detailed analyses of its attention distribution matrix.

## 2 RELATED WORKS

**Transformer-Based Time Series Modeling**. Transformer-based models have significantly advanced time series modeling by capturing long-range dependencies through sophisticated attention mechanisms (Vaswani et al., 2017). Recent efforts have explored diverse improvements in time series forecasting, particularly in computational efficiency and temporal periodicity modeling. For efficiency, Informer introduced ProbSparse attention to process long sequences with reduced computational demands (Zhou et al., 2021), followed by Pyraformer, which employs pyramidal attention to capture multi-scale dependencies efficiently (Liu et al., 2022). Triformer further optimizes memory usage with triangle attention (Cirstea et al., 2022), and PatchTST leverages patch-based processing to minimize resource requirements (Nie et al., 2023). To capture periodicity, Autoformer integrates auto-correlation to model recurring patterns effectively (Wu et al., 2021), while FEDformer enhances accuracy by combining frequency-domain attention with decomposition (Zhou et al., 2022). Despite these advancements, the opaque internal workings of Transformers and redundant linear projections often limit their interpretability and efficiency (Zeng et al., 2023). More recently, PFformer introduces position-free embeddings to better capture complex dependencies in extreme adaptive multivariate settings (Li & Anastasiu, 2025), and PENGUIN leverages periodic-nested group attention with relative bias to explicitly model multiple coexisting temporal periodicities (Sun et al., 2025). This work addresses these limitations by reinterpreting Transformers through a graph-based lens, enabling clearer mechanistic insights and streamlined computations.

**Graph-Based Methods and Interpretability**. Graph Convolutional Networks (GCNs) effectively capture graph-structured data by aggregating features across multi-hop neighborhoods using static adjacency matrices (Kipf & Welling, 2017; Wu et al., 2019; Yu et al., 2018). In multivariate time series forecasting, models such as GTS (Shang et al., 2021), MTGNN (Wu et al., 2020), AGCRN (Bai et al., 2020), and StemGNN (Cao et al., 2020) typically construct (static or adaptive) graphs over the channel dimension and handle temporal dynamics separately via convolutions or recurrence,

---

[1] A feed-forward network is a 2-layer multilayer perceptron.

leaving the temporal axis itself unmodeled as a graph. FourierGNN (Yi et al., 2023) stands out as a rare exception by constructing a joint space-time graph over both channels and time steps, yet its fully connected design incurs prohibitive complexity, necessitating sliding windows that severely constrain long-range temporal modeling. Some studies have begun treating sequences as graphs (Li et al., 2024; Joshi, 2025), yet they lack a systematic theoretical comparison of Transformers and GCNs across both forward and backward passes, fail to eliminate redundant linear projections, and do not effectively leverage explicit multi-hop aggregation. Existing interpretability efforts in time series, such as attention visualization (Vaswani et al., 2017) or feature attribution methods (Wickstrøm et al., 2020), are largely post-hoc and fail to provide intrinsic mechanistic insights. In contrast, our Fighter architecture bridges Transformers and GCNs by casting the attention distribution matrix as a dynamic, learnable adjacency matrix defined exclusively over the temporal dimension, removes redundant linear projections for streamlined computation, and incorporates multi-hop aggregation to explicitly represent temporal dependencies as interpretable graph edges, significantly enhancing both efficiency and interpretability.

## 3 BACKGROUND

**Notation.**[2] Consider a graph $G = (X_{n \times d}, A_{n \times n}) \in \mathcal{G}$, where $X_{n \times d}$ is an $n \times d$ feature matrix comprising all node feature vectors $x_i \in \mathbb{R}^d$, for $i \in \mathbb{N}_n$ with $\mathbb{N}_n := \{1, \ldots, n\}$, and $A_{n \times n}$ is the adjacency matrix encoding the graph's edge connections. Additionally, let $S_{1:S} = (x_1, \ldots, x_S) \in \mathcal{S}$ denote a sequence of length $S$, where each $x_s \in \mathbb{R}^d$ is the feature vector for the $s$-th element, $s \in \mathbb{N}_S$, and the collection of these vectors forms an $S \times d$ feature matrix, denoted $X_{S \times d}$. In both graphs and sequences, feature vectors $x$ are row vectors, written as $[x_j]_d^\top = [x_1, \ldots, x_d]$. The $i$-th row and $j$-th column of a feature matrix, corresponding to the $i$-th node or element and $j$-th feature, are denoted by $X_{(i,:)} := e_i^\top X$ and $X_{(:,j)} := X e_j$, respectively, where $e_i$ is a standard basis vector with a 1 in the $i$-th position and 0 elsewhere. The identity matrix is denoted by $I$. A block diagonal matrix with entry matrices $M_1, \ldots, M_m$ is denoted $\mathrm{blkdiag}(M_1, \ldots, M_m)$, and if all $m$ entries are identical, it is simplified to $\mathrm{blkdiag}(M; m)$.

**Flexible graph convolutional network (GCN)** extends traditional GCN by enabling adaptable weight assignments for aggregated features (Krishnagopal & Ruiz, 2023). Unlike conventional GCN, which uses $(A + I)^\kappa X$ for feature aggregation—limiting flexibility by applying uniform weights to features from different hops (*i.e.*, neighbor distances) within a single layer $\sigma((A + I)^\kappa X W)$ with weight matrix $W$—a flexible GCN $X_{\mathrm{GCN}}^{(L)}$ unfolds these features and assigns distinct weights (Abu-El-Haija et al., 2019; Zhang et al., 2025). This can be expressed as:

$$X_{\mathrm{GCN}}^{(\ell)} = \sigma\left(A^{[\kappa_\ell]} \mathrm{blkdiag}(X_{\mathrm{GCN}}^{(\ell-1)}; \kappa_\ell) \cdot W^{(\ell)}\right), \ell \in \mathbb{N}_{L-1}$$

$$X_{\mathrm{GCN}}^{(L)} = A^{[\kappa_L]} \mathrm{blkdiag}(X_{\mathrm{GCN}}^{(L-1)}; \kappa_L) \cdot W^{(L)}, \tag{1}$$

where $A^{[\kappa]} := \|_{i=0}^{\kappa-1} A^i = [I, A, \cdots, A^{\kappa-1}]$ is an $n \times \kappa n$ matrix formed by concatenation $\|$, $W^{(\ell)}$ is the weight matrix at layer $\ell$ with dimensions $h_{\ell-1} \times h_\ell$, $h_\ell$ represents the width of layer $\ell$ with $h_0 = d$, and $\kappa_\ell$ denotes the hop distance at the $\ell$-th layer. $X_{\mathrm{GCN}}^{(0)}$ is the input feature matrix, and $\sigma$ denotes the activation function (*e.g.*, ReLU). Such a flexible GCN resembles an MLP but aggregates features across multiple hops at the start of each layer. By restricting the multi-hop aggregation, governed by $A^{[\kappa_\ell]} \mathrm{blkdiag}(X_{\mathrm{GCN}}^{(\ell-1)}; \kappa_\ell)$, to single-hop aggregation (*i.e.*, using only the $A^1$ term), the feature representation $X_{\mathrm{GCN}}^{(\ell)}$ at layer $\ell \in \mathbb{N}_{L-1}$ is simplified as:

$$X_{\mathrm{GCN}}^{(\ell)} = \sigma\left(A X_{\mathrm{GCN}}^{(\ell-1)} W^{(\ell)}\right), \ell \in \mathbb{N}_{L-1}. \tag{2}$$

**Transformer**, originally developed for natural language processing, comprises two main components: an encoder and a decoder (Vaswani et al., 2017). For representation learning in time series modeling, the Transformer encoder $X_{\mathrm{TF}}^{(L)}$, with its attention layer $X_{\mathrm{Att}}^{(\ell)}$ and FFN layer $X_{\mathrm{FFN}}^{(\ell)}$, has demonstrated high effectiveness (Zerveas et al., 2021; Yıldız et al., 2022). Specifically, the Transformer encoder

---

[2]See the notation table in Appendix A.1.

with self-attention can be formulated as:

$$\boldsymbol{X}_{\text{Att}}^{(\ell)} = \text{smx}\left(d^{-1/2}\boldsymbol{X}_{\text{FFN}}^{(\ell-1)}\boldsymbol{W}_Q^{(\ell)}\left(\boldsymbol{X}_{\text{FFN}}^{(\ell-1)}\boldsymbol{W}_K^{(\ell)}\right)^\top\right)\boldsymbol{X}_{\text{FFN}}^{(\ell-1)}\boldsymbol{W}_V^{(\ell)}, \ell \in \mathbb{N}_{L-1}$$

$$\boldsymbol{X}_{\text{FFN}}^{(\ell)} = \sigma\left(\boldsymbol{X}_{\text{Att}}^{(\ell)}\boldsymbol{W}_{\text{FFN}}^{(\ell,1)} + \boldsymbol{b}_{\text{FFN}}^{(\ell,1)}\right)\boldsymbol{W}_{\text{FFN}}^{(\ell,2)} + \boldsymbol{b}_{\text{FFN}}^{(\ell,2)}$$

$$\boldsymbol{X}_{\text{TF}}^{(L)} = \boldsymbol{X}_{\text{FFN}}^{(L-1)}\boldsymbol{W}^{(L)}, \tag{3}$$

where $\text{smx}(\cdot)$ denotes a row-wise softmax operation; $\boldsymbol{W}_Q^{(\ell)}$, $\boldsymbol{W}_K^{(\ell)}$, and $\boldsymbol{W}_V^{(\ell)}$ are the query, key, and value weight matrices at layer $\ell$, with dimensions $h_{\ell-1} \times p$, $h_{\ell-1} \times p$, $h_{\ell-1} \times v$ respectively; $\boldsymbol{W}_{\text{FFN}}^{(\ell,1)}$ and $\boldsymbol{W}_{\text{FFN}}^{(\ell,2)}$ are the weight matrices for the first and second sublayers of the FFN at layer $\ell$, with dimension $v \times h_\ell$ and $h_\ell \times h_\ell$; $\boldsymbol{b}_{\text{FFN}}^{(\ell,\cdot)}$ represents the bias terms; $\boldsymbol{X}_{\text{FFN}}^{(0)}$ is the input feature matrix of size $S \times d$ with $d = h_0$; and $\sigma$ is the row-wise activation function (*e.g.*, ReLU). For clarity and conciseness, this paper discusses a single-head Transformer encoder, omitting tokenization, positional encodings, layer normalization, and residual connections from the formulation. These components are fully incorporated in our experimental implementation.

## 4 TRANSFORMERS AS DYNAMIC GRAPH CONVOLUTION FOR TIME SERIES

We begin by analyzing the forward pass of Transformer encoders, where the attention distribution matrix serves as a dynamic adjacency matrix, and its composition with subsequent transformations performs graph convolution-like computations. Extending this analysis to the training phase, we show that gradients update the value matrix and FFN parameters similarly to GCNs, while the query and key weight matrices enable adaptive updates to a learnable adjacency matrix. Building on these findings, we introduce Fighter, a flexible graph convolutional transformer that removes redundant linear projections and incorporates multi-hop graph aggregation for interpretable modeling of temporal dependencies.

### 4.1 FORWARD PASS: THE ATTENTION DISTRIBUTION MATRIX AS A DYNAMIC ADJACENCY

To reveal the graph convolutional nature of Transformers, we reformulate the Transformer encoder and compare it to GCNs. For a Transformer encoder as expressed in Equation 3, the learned hidden feature $\boldsymbol{X}_{\text{TF}}^{(\ell)}$ at layer $\ell \in \mathbb{N}_{L-1}$ can be reorganized, omitting bias terms for simplicity and defining $\boldsymbol{W}_V^{(\ell,1)} := \boldsymbol{W}_V^{(\ell)}\boldsymbol{W}_{\text{FFN}}^{(\ell,1)}$, as:

$$\boldsymbol{X}_{\text{TF}}^{(\ell)} = \sigma\left(\text{smx}\left(d^{-1/2}\boldsymbol{X}_{\text{TF}}^{(\ell-1)}\boldsymbol{W}_Q^{(\ell)}\left(\boldsymbol{X}_{\text{TF}}^{(\ell-1)}\boldsymbol{W}_K^{(\ell)}\right)^\top\right)\boldsymbol{X}_{\text{TF}}^{(\ell-1)}\boldsymbol{W}_V^{(\ell,1)}\right) \cdot \boldsymbol{W}_{\text{FFN}}^{(\ell,2)}. \tag{4}$$

The term $\text{smx}\left(d^{-1/2}\boldsymbol{X}_{\text{TF}}^{(\ell-1)}\boldsymbol{W}_Q^{(\ell)}(\boldsymbol{X}_{\text{TF}}^{(\ell-1)}\boldsymbol{W}_K^{(\ell)})^\top\right)$ forms an $S \times S$ attention distribution matrix, which performs single-hop feature aggregation over the input sequence, analogous to the adjacency matrix $\boldsymbol{A}$ in GCNs (Veličković et al., 2018). The derivation of this reformulation is provided in Appendix A.2.

By comparing Equations 4 and 2, we observe that the attention distribution matrix in the Transformer serves as a dynamic analogue to the static adjacency matrix $\boldsymbol{A}$ in GCNs, while $\boldsymbol{W}_V^{(\ell,1)}$ aligns with $\boldsymbol{W}^{(\ell)}$ for feature extraction. That is to say, a Transformer encoder, to some extent, is a single-hop GCN with dynamic adjacency matrix. From this GCN perspective, the additional linear projection $\boldsymbol{W}_{\text{FFN}}^{(\ell,2)}$ in the Transformer appears redundant, as GCNs do not employ a post-activation linear projection. Indeed, its role in transforming aggregated features is effectively subsumed by the subsequent layer's projections, namely $\boldsymbol{W}_Q^{(\ell+1)}$, $\boldsymbol{W}_K^{(\ell+1)}$, and $\boldsymbol{W}_V^{(\ell+1,1)}$, which collectively manage feature transformations in the next iteration of the recursive architecture. This insight highlights a structural parallel between Transformers and GCNs that has been underexplored. In contrast, prior works (Veličković et al., 2018; Li et al., 2024; Joshi, 2025) focus predominantly on the attention mechanism's role in isolation, often missing its broader graph convolutional interpretation.

For the final layer, $\ell = L$, the Transformer encoder omits the attention mechanism, yielding $\boldsymbol{X}_{\text{TF}}^{(L)} = \boldsymbol{X}_{\text{TF}}^{(L-1)}\boldsymbol{W}^{(L)}$. This directly corresponds to the GCN formulation when $\kappa_L = 1$, *i.e.*, $\boldsymbol{X}_{\text{GCN}}^{(L)} =$

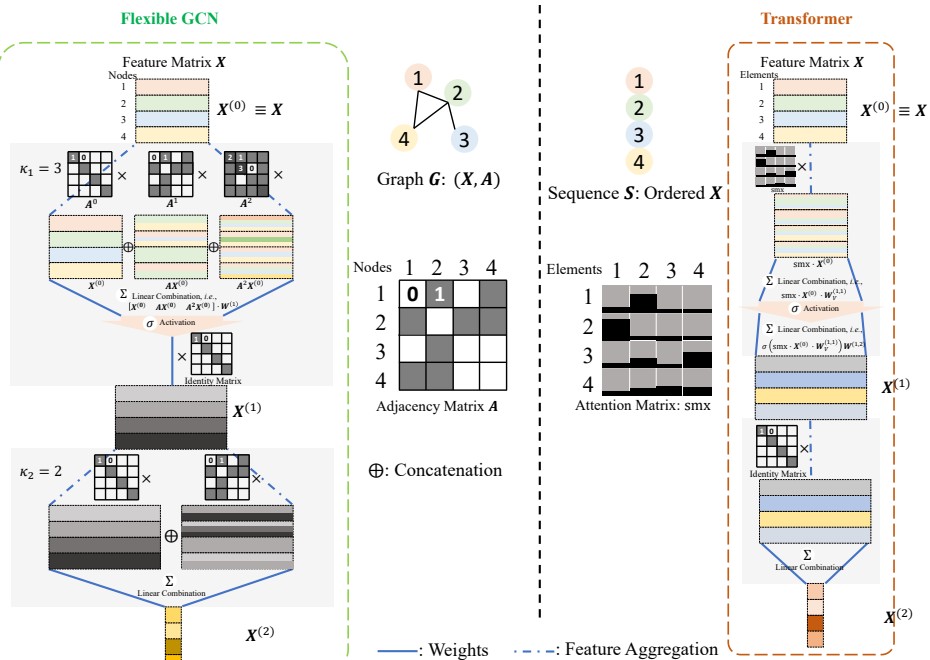

Figure 1: Visual comparison of a Transformer encoder and a flexible GCN, illustrating the equivalence between the Transformer's attention distribution matrix and the GCN's adjacency matrix in performing feature aggregation during the forward pass.

$A^0 X_{\text{GCN}}^{(L-1)} W^{(L)}$ (Li et al., 2024). This equivalence underscores that Transformers inherently perform graph-like convolution during the forward pass, with the attention distribution matrix dynamically adapting to the input sequence. A visual comparison of the Transformer encoder and flexible GCN is provided in Figure 1.

**Remark 1.** *The equivalence between Transformer encoders and GCNs stems from the attention distribution matrix dynamically mimicking the GCN's adjacency matrix for feature aggregation. This equivalence is independent of the specific attention mechanism employed, as the operation within* smx *can be tailored to the chosen mechanism (e.g., scaled dot-product or variants). This flexibility ensures the graph convolutional interpretation remains applicable across diverse Transformer architectures, unifying sequence and graph-based processing under a common framework.*

## 4.2 Backward Pass: Gradient-Based Learning of Dynamic Structure

Following the graph convolutional interpretation of Transformer encoders in the above Section, we now explore the backward pass to compare the gradient-based learning dynamics of Transformers encoders and GCNs (Gelfand et al., 2000; Ruder, 2016; Chami et al., 2022). We demonstrate that the feature transformation parameters in Transformers, excluding the query and key weight matrices $W_Q$ and $W_K$, are updated similarly to those in GCNs. Additionally, we show how the attention distribution matrix, analogous to the GCN's adjacency matrix, is dynamically learned through the gradients of $W_Q$ and $W_K$. For clarity, we consider a two-layer, scalar-valued Transformer encoder and a GCN, as illustrated in Figure 1.

For the Transformer encoder, we simplify Equation 4 by omitting the redundant linear projection $W_{\text{FFN}}^{(\ell,2)}$ and setting $L = 2$, yielding the output at the final layer:

$$X_{\text{TF}}^{(2)} = \sigma\left(\text{smx}\left(d^{-1/2} X^{(0)} W_Q^{(1)} \left(X^{(0)} W_K^{(1)}\right)^\top\right) X^{(0)} W_V^{(1,1)}\right) \cdot W^{(2)}. \tag{5}$$

The gradients with respect to the feature transformation parameters $\boldsymbol{W}^{(2)}$ and $\boldsymbol{W}_{V(:,i)}^{(1,1)}$ ($i \in \mathbb{N}_{h_1}$) are:

$$
\begin{aligned}
\frac{\partial \boldsymbol{X}_{\mathrm{TF}}^{(2)}}{\partial \boldsymbol{W}^{(2)}} &= \sigma\bigg(\mathrm{smx}\Big(d^{-1/2}\boldsymbol{X}^{(0)}\boldsymbol{W}_Q^{(1)}\big(\boldsymbol{X}^{(0)}\boldsymbol{W}_K^{(1)}\big)^{\top}\Big)\boldsymbol{X}^{(0)}\boldsymbol{W}_V^{(1,1)}\bigg), \\
\frac{\partial \boldsymbol{X}_{\mathrm{TF}}^{(2)}}{\partial \boldsymbol{W}_{V(:,i)}^{(1,1)}} &= \dot{\sigma} \cdot \mathrm{smx}\Big(d^{-1/2}\boldsymbol{X}^{(0)}\boldsymbol{W}_Q^{(1)}\big(\boldsymbol{X}^{(0)}\boldsymbol{W}_K^{(1)}\big)^{\top}\Big)\boldsymbol{X}^{(0)} \cdot \boldsymbol{W}_{(i,:)}^{(2)},
\end{aligned}
\tag{6}
$$

where $\dot{\sigma}$ is the derivative of the activation function.

For the GCN, we consider single-hop aggregation in the first layer and set $\kappa_2 = 1$ in the second layer, yielding the gradients for Equation 2:

$$
\frac{\partial \boldsymbol{X}_{\mathrm{GCN}}^{(2)}}{\partial \boldsymbol{W}^{(2)}} = \sigma(\boldsymbol{A}\boldsymbol{X}^{(0)}\boldsymbol{W}^{(1)}), \qquad \frac{\partial \boldsymbol{X}_{\mathrm{GCN}}^{(2)}}{\partial \boldsymbol{W}_{(:,i)}^{(1)}} = \dot{\sigma}\boldsymbol{A}\boldsymbol{X}^{(0)}\boldsymbol{W}_{(i,:)}^{(2)},
\tag{7}
$$

which correspond to a specific instance of the multi-hop gradient, with details provided in Appendix A.3.

Comparing Equations 6 and 7, the gradients for feature transformation parameters in both models share a similar structure, with the Transformer's attention distribution matrix $\mathrm{smx}\Big(d^{-1/2}\boldsymbol{X}^{(0)}\boldsymbol{W}_Q^{(1)}\big(\boldsymbol{X}^{(0)}\boldsymbol{W}_K^{(1)}\big)^{\top}\Big)$ acting as a dynamic counterpart to the GCN's static adjacency matrix $\boldsymbol{A}$. This similarity indicates that the Transformer's feature transformation parameters are updated in a manner analogous to those in GCNs.

However, the Transformer's attention distribution matrix is dynamically learned through the gradients of $\boldsymbol{W}_Q^{(1)}$ and $\boldsymbol{W}_K^{(1)}$ ($i \in \mathbb{N}_p$):

$$
\frac{\partial \boldsymbol{X}_{\mathrm{TF}}^{(2)}}{\partial \boldsymbol{W}_{Q(:,i)}^{(1)}} = \Big[\dot{\sigma}_j/\sqrt{d}\,\underbrace{\boldsymbol{X}_{(j,:)}^{(0)}}_{1\times d} \cdot \underbrace{\Big(\overbrace{\mathcal{K}_{(:,i)}^{(1)}}^{1\times S}{}^{\top}\overbrace{\mathrm{blkdiag}\big(\mathrm{smx}(\xi_{(j,i::,i)}^{(1)})\big)}^{S\times S}\overbrace{\boldsymbol{X}^{(0)}}^{S\times d}\overbrace{\boldsymbol{W}_V^{(1,1)}}^{d\times h_1}\overbrace{\boldsymbol{W}^{(2)}}^{h_1\times 1} - \overbrace{\mathcal{K}_{(:,i)}^{(1)}}^{1\times S}{}^{\top}\overbrace{(\mathrm{smx}(\xi_{(j,i::,i)}^{(1)}))^{\top}\mathrm{smx}(\xi_{(j,i::,i)}^{(1)})}^{S\times S}\overbrace{\boldsymbol{X}^{(0)}}^{S\times d}\overbrace{\boldsymbol{W}_V^{(1,1)}}^{d\times h_1}\overbrace{\boldsymbol{W}^{(2)}}^{h_1\times 1}\Big)}_{:=\omega_j,\,1\times 1}\Big]_{S\times d} \tag{8}
$$

$$
\frac{\partial \boldsymbol{X}_{\mathrm{TF}}^{(2)}}{\partial \boldsymbol{W}_{K(:,i)}^{(1)}} = \Big[\dot{\sigma}_j/\sqrt{d}\,\underbrace{\boldsymbol{X}_{(j,:)}^{(0)}}_{1\times d} \cdot \underbrace{\Big(\overbrace{\mathcal{Q}_{(:,i)}^{(1)}}^{1\times S}{}^{\top}\overbrace{\mathrm{blkdiag}\big(\mathrm{smx}(\xi_{(j,i::,i)}^{(1)})\big)}^{S\times S}\overbrace{\boldsymbol{X}^{(0)}}^{S\times d}\overbrace{\boldsymbol{W}_V^{(1,1)}}^{d\times h_1}\overbrace{\boldsymbol{W}^{(2)}}^{h_1\times 1} - \overbrace{\mathcal{Q}_{(:,i)}^{(1)}}^{1\times S}{}^{\top}\overbrace{(\mathrm{smx}(\xi_{(j,i::,i)}^{(1)}))^{\top}\mathrm{smx}(\xi_{(j,i::,i)}^{(1)})}^{S\times S}\overbrace{\boldsymbol{X}^{(0)}}^{S\times d}\overbrace{\boldsymbol{W}_V^{(1,1)}}^{d\times h_1}\overbrace{\boldsymbol{W}^{(2)}}^{h_1\times 1}\Big)}_{1\times 1}\Big]_{S\times d}, \tag{9}
$$

where $\mathcal{Q} := \boldsymbol{X}^{(0)}\boldsymbol{W}_Q^{(1)}, \mathcal{K} := \boldsymbol{X}^{(0)}\boldsymbol{W}_K^{(1)}, \xi_{(j,i::,i)}^{(1)} := \mathcal{Q}_{(j,i)}^{(1)}\mathcal{K}_{(:,i)}^{(1)}{}^{\top}/\sqrt{d}$, and the derivation is provided in Appendix A.4. Focusing on the query gradient (Equation 8), the update depends on both the input features $\boldsymbol{X}_{(j,:)}^{(0)}$ and a scalar $\omega_j$, which assigns varying importance to each sequence element during training. This dynamic adaptation of the attention distribution matrix, absent in the fixed adjacency matrix of GCNs, interprets the Transformer's expressiveness.

## 4.3 FIGHTER: EFFICIENT MULTI-HOP GRAPH CONVOLUTIONAL TRANSFORMER

Building on the unified theoretical reinterpretation of Transformer encoders as GCNs, we propose **Fighter** (Flexible Graph Convolutional Transformer), a streamlined architecture that eliminates redundant linear projections and incorporates multi-hop graph aggregation. This design enhances expressive power by extending the dynamic adjacency matrix's capabilities in the forward pass while reducing computational overhead. Specifically, the forward pass of Fighter for the learned hidden features $\boldsymbol{X}_{\mathrm{FT}}^{(\ell)}$ at layer $\ell \in \mathbb{N}_{L-1}$ and the final output $\boldsymbol{X}_{\mathrm{FT}}^{(L)}$ is expressed as:

$$
\boldsymbol{X}_{\mathrm{FT}}^{(\ell)} = \sigma\left(\mathrm{smx}^{[\kappa_\ell]}\left(d^{-1/2}\boldsymbol{X}_{\mathrm{FT}}^{(\ell-1)}\boldsymbol{W}_Q^{(\ell)}\left(\boldsymbol{X}_{\mathrm{FT}}^{(\ell-1)}\boldsymbol{W}_K^{(\ell)}\right)^{\top}\right)\mathrm{blkdiag}(\boldsymbol{X}_{\mathrm{FT}}^{(\ell-1)};\kappa_\ell)\cdot\boldsymbol{W}^{(\ell)}\right)
$$

$$
\boldsymbol{X}_{\mathrm{FT}}^{(L)} = \mathrm{smx}^{[\kappa_L]}\left(d^{-1/2}\boldsymbol{X}_{\mathrm{FT}}^{(L-1)}\boldsymbol{W}_Q^{(L)}\left(\boldsymbol{X}_{\mathrm{FT}}^{(L-1)}\boldsymbol{W}_K^{(L)}\right)^{\top}\right)\mathrm{blkdiag}(\boldsymbol{X}_{\mathrm{FT}}^{(L-1)};\kappa_L)\cdot\boldsymbol{W}^{(L)} \tag{10}
$$

where the value and feed-forward projection is streamlined, and multi-hop aggregation is enabled through the raised attention distribution matrix $\mathrm{smx}^{[\kappa_\ell]}$ and the block-diagonal replication $\mathrm{blkdiag}(\cdot;\kappa_\ell)$. A visualization of Fighter is provided in Figure 2.

Fighter reinterprets the Transformer encoder through the lens of a GCN with a dynamic adjacency matrix, as established in the forward pass analysis (Section 4.1). Here, the attention distribution matrix $\mathrm{smx}\left(d^{-1/2}\boldsymbol{X}_{\mathrm{TF}}^{(\ell-1)}\boldsymbol{W}_Q^{(\ell)}\left(\boldsymbol{X}_{\mathrm{TF}}^{(\ell-1)}\boldsymbol{W}_K^{(\ell)}\right)^{\top}\right)$ functions as a learnable counterpart to the static adjacency matrix $\boldsymbol{A}$ in GCNs. To mitigate the redundancy of the additional linear projection $\boldsymbol{W}_{\mathrm{FFN}}^{(\ell,2)}$—which, as shown, is effectively subsumed by subsequent layer projections—Fighter removes it entirely. Instead, it relies on the combined value and initial FFN projection (consolidated into $\boldsymbol{W}^{(\ell)}$) alongside transformations in ensuing layers for efficient feature extraction, thereby simplifying the architecture without sacrificing performance.

Furthermore, to overcome the limitations of single-hop aggregation in standard Transformers (Equation 4), Fighter integrates multi-hop aggregation, drawing inspiration from the flexible GCN (Equation 1). By raising the attention distribution matrix to powers up to $\kappa_\ell - 1$ and replicating input features accordingly, Fighter explicitly captures multi-scale temporal dependencies. This models indirect relationships across time steps as multi-hop graph connections, enabling Fighter to better handle complex, long-range patterns inherent in time series data.

In the backward pass, Fighter retains the dynamic learning of the adjacency matrix through gradients with respect to $\boldsymbol{W}_Q^{(\ell)}$ and $\boldsymbol{W}_K^{(\ell)}$, as analyzed in Section 4.2. These updates adapt the attention distribution based on input features and their contextual importance, providing an adaptive refinement of temporal relationships during training—a capability absent in static GCNs. This combination of streamlined design and enhanced aggregation yields an efficient, interpretable model that bridges sequence and graph paradigms for superior time series modeling.

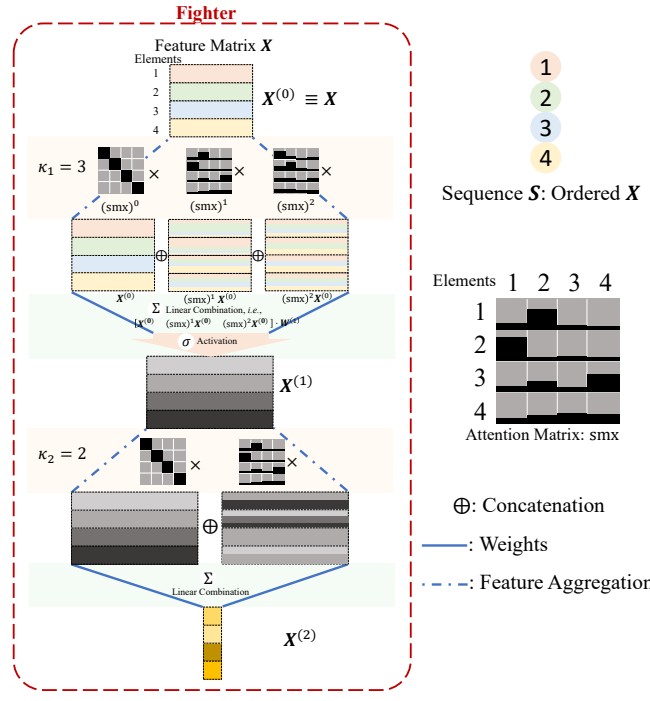

Figure 2: Visual illustration of the Fighter architecture, showcasing its streamlined design with multi-hop graph aggregation and dynamic adjacency matrix for efficient time series modeling.

**Remark 2.** *Although Equation 10 presents a simplified formulation without multi-head attention, residual connections, or layer normalization for clarity, Fighter is designed as a modular plug-in that can seamlessly integrate these components into any attention-based framework. In our experimental implementation, we adopt these standard enhancements to stabilize training and improve performance, as is common in Transformer architectures. The Pseudocode 1 shows Fighter's core forward pass, with optional lines for incorporating multi-head attention (via parallel heads and concatenation), residuals, and layer normalization.*

## 5 EXPERIMENTS AND RESULTS

In our experiments, we evaluate the performance of Fighter on time series forecasting tasks to validate its effectiveness in capturing temporal dependencies. The overall experimental results are presented in Table 1, which clearly highlights the superiority of Fighter over standard Transformer-based baselines including Autoformer (Wu et al., 2021), Informer(Zhou et al., 2021), Reformer (Kitaev et al., 2020), and vanilla Transformer (Vaswani et al., 2017). Furthermore, Fighter exhibits versatility in text sequence modeling, achieving competitive results with a streamlined architecture that enhances

Table 1: Time series forecasting (MSE/MAE) results on benchmark datasets and text sequence classification accuracy (Acc) on the AG News dataset. 'O' denotes the prediction length. Lower MSE/MAE indicates better forecasting performance, while higher accuracy reflects superior classification. The best results are highlighted in bold.

| Dataset | O | Autoformer | | Informer | | Reformer | | Transformer | | Fighter ($\kappa = 3$) | |
|---|---|---|---|---|---|---|---|---|---|---|---|
| | | MSE | MAE | MSE | MAE | MSE | MAE | MSE | MAE | MSE | MAE |
| Electricity | 96 | 0.474±0.078 | 0.485±0.041 | 0.544±0.006 | 0.546±0.003 | 0.374±0.004 | 0.438±0.004 | 0.388±0.002 | 0.442±0.002 | **0.339±0.007** | **0.417±0.004** |
| | 192 | 0.581±0.157 | 0.533±0.074 | 0.652±0.062 | 0.616±0.032 | 0.387±0.010 | 0.447±0.007 | 0.384±0.019 | **0.441±0.013** | **0.376±0.034** | 0.448±0.027 |
| | 336 | 0.463±0.078 | 0.484±0.038 | 0.686±0.015 | 0.632±0.005 | 0.466±0.013 | 0.502±0.009 | 0.441±0.053 | 0.487±0.042 | **0.408±0.015** | **0.477±0.012** |
| | 720 | 0.559±0.194 | 0.539±0.105 | 0.966±0.007 | 0.801±0.004 | 0.490±0.014 | 0.519±0.008 | 0.442±0.045 | 0.484±0.033 | **0.380±0.010** | **0.451±0.007** |
| Weather | 96 | 0.392±0.023 | 0.444±0.013 | 0.911±0.095 | 0.709±0.039 | 1.076±0.129 | 0.813±0.054 | 0.564±0.005 | 0.545±0.004 | **0.270±0.008** | **0.344±0.007** |
| | 192 | 0.348±0.023 | 0.415±0.020 | 0.898±0.036 | 0.708±0.017 | 1.080±0.093 | 0.806±0.041 | 0.620±0.013 | 0.569±0.010 | **0.256±0.012** | **0.330±0.009** |
| | 336 | 0.384±0.037 | 0.430±0.028 | 0.930±0.020 | 0.729±0.010 | 1.038±0.019 | 0.788±0.002 | 0.669±0.102 | 0.585±0.050 | **0.296±0.008** | **0.358±0.007** |
| | 720 | 0.628±0.040 | 0.587±0.027 | 1.137±0.025 | 0.831±0.014 | 0.693±0.137 | 0.614±0.073 | 0.957±0.078 | 0.724±0.035 | **0.342±0.004** | **0.383±0.004** |
| ETTh1 | 96 | 1.117±0.043 | 0.846±0.010 | 1.135±0.049 | 0.799±0.016 | 1.139±0.016 | 0.791±0.009 | 0.919±0.040 | 0.754±0.022 | **0.732±0.076** | **0.651±0.050** |
| | 192 | 1.148±0.085 | 0.839±0.026 | 1.108±0.060 | 0.793±0.028 | 1.251±0.035 | 0.840±0.014 | 0.949±0.016 | 0.762±0.007 | **0.727±0.300** | **0.631±0.164** |
| | 336 | 1.156±0.031 | 0.867±0.006 | 1.246±0.033 | 0.883±0.019 | 1.383±0.016 | 0.884±0.004 | 0.971±0.058 | 0.778±0.031 | **0.912±0.242** | **0.759±0.135** |
| | 720 | 1.108±0.044 | 0.846±0.019 | 1.267±0.019 | 0.873±0.010 | 1.425±0.009 | 0.894±0.003 | 1.049±0.007 | 0.810±0.009 | **0.897±0.163** | **0.763±0.096** |
| ETTm1 | 96 | 0.903±0.004 | 0.736±0.007 | 0.916±0.044 | 0.718±0.027 | 1.034±0.015 | 0.744±0.009 | 0.572±0.046 | 0.542±0.024 | **0.500±0.032** | **0.499±0.021** |
| | 192 | 0.877±0.078 | 0.728±0.046 | 0.955±0.029 | 0.727±0.014 | 1.196±0.017 | 0.807±0.005 | 0.690±0.046 | 0.609±0.018 | **0.517±0.036** | **0.514±0.034** |
| | 336 | 0.999±0.014 | 0.788±0.031 | 1.046±0.068 | 0.771±0.031 | 1.391±0.018 | 0.885±0.002 | 0.807±0.014 | 0.675±0.010 | **0.603±0.066** | **0.572±0.041** |
| | 720 | 0.998±0.105 | 0.787±0.051 | 1.090±0.022 | 0.795±0.005 | 1.538±0.001 | 0.926±0.001 | 1.002±0.030 | 0.751±0.003 | **0.701±0.026** | **0.627±0.029** |
| | | Acc | | Acc | | Acc | | Acc | | Acc | |
| AG News | – | 0.876 | | 0.831 | | 0.820 | | 0.864 | | **0.898** | |

efficiency compared to traditional Transformer-based baselines. Detailed settings and additional discussion are given in Appendix B.

We assess Fighter's performance using widely recognized datasets for time series forecasting and text classification to ensure a comprehensive evaluation. These datasets are:

- Electricity[3]: containing the hourly electricity consumption of 321 customers from 2012 to 2014;

- Weather[4]: containing 21 meteorological indicators, recorded every 10 minutes for the 2020 year;

- ETT(Zhou et al., 2021): containing 7 electricity transformer factors, ETTh1 is recorded every hour, and ETTm1 is recorded every 15 minutes from 2016 to 2018;

- AG News(Zhang et al., 2015): containing 120,000 samples, with 30,000 instances per class, consisting of news articles from four categories: World, Sports, Business, and SCI / Technology.

Fighter extends the vanilla Transformer by incorporating higher-order compositions of the attention distribution matrix, controlled by the parameter $\kappa$, which governs the power of attention distributions. This mechanism enables **multi-hop graph aggregation**, explicitly modeling temporal dependencies across multiple scales as dynamic adjacency matrix, where dependencies are represented as graph edges. For the benchmark time series datasets, Fighter consistently outperforms most baselines, including Autoformer (Wu et al., 2021), Informer (Zhou et al., 2021), Reformer (Kitaev et al., 2020), and the vanilla Transformer (Vaswani et al., 2017) across different prediction lengths varying from 96 to 720. On the Electricity dataset, Fighter achieves the lowest MSE across all prediction horizons (96, 192, 336, and 720). Even at the longest horizon of 720, it records an MSE of 0.380, noticeably outperforming Transformer (0.442) and Reformer (0.490). At the horizon of 336, Fighter maintains a clear edge with an MSE of 0.408 compared with 0.441 (Transformer) and 0.466 (Reformer), while its MAE follows a similarly favorable trend, reflecting stable and reliable forecasting capability.

On the Weather dataset, Fighter exhibits particularly impressive results. For a prediction length of 96, it obtains an MSE/MAE of 0.270/0.344, which corresponds to a 74.9% lower MSE and 57.7% lower MAE than Reformer (1.076/0.813), and a reduction of approximately 31% and 23% relative to Autoformer (0.392/0.444). This advantageous gap persists across longer horizons; at $O = 720$, Fighter still achieves a competitive 0.342/0.383, remaining comfortably ahead of the other baselines and demonstrating remarkable robustness over extended sequences.

Fighter continues to perform favorably on the ETTh1 and ETTm1 datasets. At the longest tested horizon of 720 on ETTh1, it attains an MSE/MAE of 0.897/0.763, representing improvements of 37.1% and 14.6% over Reformer and 29.2% and 12.6% over Informer. The gains are even more pronounced on ETTm1 under the same setting, with reductions of 54.4% and 32.3% relative to Reformer, and 35.7% and 21.1% relative to Informer. Such sustained advantages as the prediction

---

[3] https://archive.ics.uci.edu/ml/datasets/ElectricityLoadDiagrams20112014
[4] https://www.bgc-jena.mpg.de/wetter

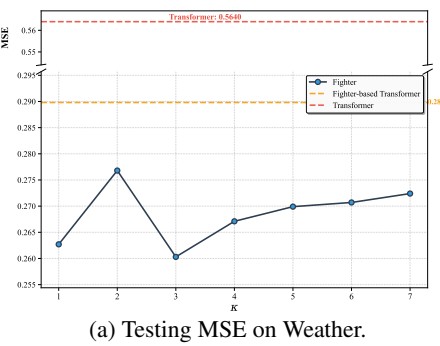 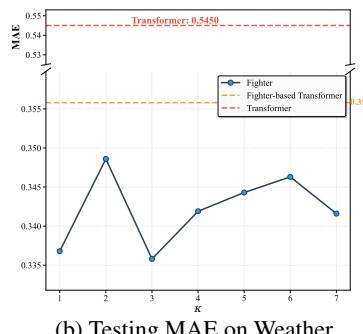

(a) Testing MSE on Weather.  (b) Testing MAE on Weather.

Figure 3: Time series forecasting performance of Fighter, Fighter-based Transformer with $\kappa \in [1, 7]$ compared to the vanilla Transformer on the Weather dataset. Fighter-based Transformer denotes implementing the Transformer using the Fighter framework. Lower MSE/MAE indicate superior performance.

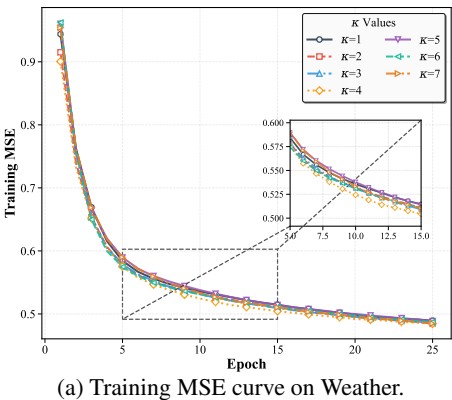 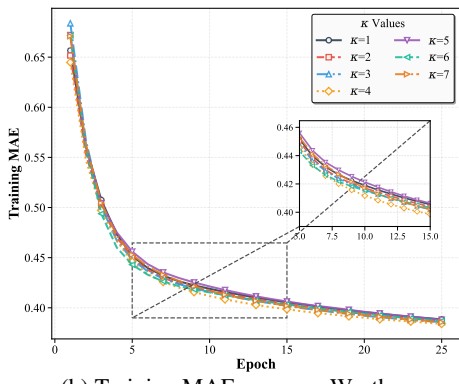

(a) Training MSE curve on Weather.  (b) Training MAE curve on Weather.

Figure 4: The training MSE/MAE performance of Fighter, with $\kappa \in [1, 7]$ on the Weather dataset. The MSE also denotes the training loss. Lower MSE/MAE indicate superior performance.

length increases highlight Fighter's effective modeling of long-range dependencies via its *multi-hop aggregation* mechanism.

Beyond time series forecasting, Fighter also shows promising versatility on the AG News text classification task, achieving 89.8% accuracy, which is higher than Transformer (86.4%), Autoformer (87.6%), and the remaining baselines, further illustrating its ability to handle diverse, high-dimensional sequential data effectively.

Fighter's multi-hop graph aggregation ensures better performance across varying prediction lengths, preventing rapid error accumulation. The interpretable nature of Fighter's attention matrix, viewed as dynamic adjacency matrix, explains its superior performance: multi-hop paths allow long-range information to propagate naturally, reducing forecasting errors and enabling robust modeling of multivariate temporal relationships. To further explore the influence of the parameter $\kappa$ on Fighter's performance, we analyze its testing MSE/MAE on the Weather dataset, as presented in Figure 3. Overall, Fighter's MSE and MAE surpass that of Fighter-based Transformer (implementing the Transformer using the Fighter framework) and the vanilla Transformer, meanwhile, Fighter and Fighter-based Transformer demonstrate better results across different $\kappa$ than the vanilla Transformer. In detail, when $\kappa$ is 3, Fighter obtains the best MSE/MAE results, which indicates Fighter's multi-hop graph aggregation effectively captures long-range attention dependencies, leading to improved feature aggregation and significantly lower errors. When $\kappa$ is 1 or 2, the dynamic attention distribution matrix captures only short-range dependencies, which is insufficient for modeling the Weather dataset's complex temporal patterns. when $\kappa$ is over 3, The MSE result remains a performance advantage over the two other baselines, indicating the sufficient capture of long-range dependencies. Beyond this point, increasing $\kappa$ results in an potential over-smoothing effect (Keriven, 2022; Rusch et al., 2023), slightly degrading performance due to excessive aggregation. A similar pattern is observed in the testing MAE, confirming the critical role of $\kappa$ in balancing dependency modeling We also present the

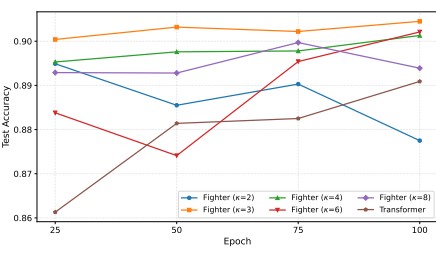 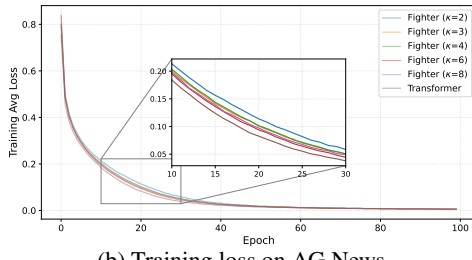

(a) Testing accuracies on AG News.  (b) Training loss on AG News.

Figure 5: Text classification performance of Fighter with $\kappa \in \{2, 3, 4, 6, 8\}$ versus the standard Transformer on the AG News dataset, evaluated across epochs 25 to 100. Higher accuracy reflects better performance.

training MSE/MAE performance comparison of Fighter across varying $\kappa$ in Figure 4. Additionally, we examine $\kappa$'s effect for text sequence modeling on the AG News dataset, as shown in Figure 5.

To further validate the effectiveness of the proposed multi-hop graph aggregation mechanism, we conduct an ablation study by incorporating it ($\kappa = 3$ applied on the attention distribution matrix) into the encoder-only iTransformer model. As shown in Table 2, integrating this mechanism consistently improves the forecasting performance of iTransformer on the ETTh1 and ETTm1 datasets across short and long horizons, with clear reductions in both MSE and MAE. These gains confirm that the multi-hop aggregation serves as a general and effective enhancement for capturing long-range dependencies, even when applied to strong state-of-the-art baselines.

Table 2: Ablation study on the multi-hop graph aggregation ($\kappa = 3$). Forecasting performance (MSE/MAE) of the encoder-only iTransformer versus iTransformer + multi-hop graph aggregation on the ETTh1 and ETTm1 datasets at prediction horizons of 96 and 720. Best results are highlighted in bold.

| Model | O | ETTh1 | | ETTm1 | |
|---|---|---|---|---|---|
| | | MSE | MAE | MSE | MAE |
| iTransformer | 96 | 0.413 | 0.423 | 0.367 | 0.388 |
| | 720 | 0.547 | 0.516 | 0.497 | 0.460 |
| iTransformer + Multi-hop ($\kappa = 3$) | 96 | **0.404** | **0.417** | **0.340** | **0.374** |
| | 720 | **0.522** | **0.505** | **0.488** | **0.454** |

## 6  CONCLUDING REMARKS AND FUTURE WORK

This work establishes a novel equivalence between Transformer encoders and Graph Convolutional Networks (GCNs), showing that the attention distribution matrix serves as a dynamic adjacency matrix, with backward pass updates to the value matrix and FFN resembling that to GCN parameters. this unified reinterpretation demystifies Transformers in time series modeling, bridging sequence and graph paradigms. Our proposed **Fighter** architecture leverages this insight by removing redundant projections and integrating multi-hop graph aggregation, offering interpretable temporal dependency representations as graph edges. Fighter achieves competitive performance on forecasting benchmarks while enhancing mechanistic interpretability.

Future work would be extending this graph-convolutional reinterpretation to other Transformer variants such as Conformers (Gulati et al., 2020) in speech processing and Diffusion Transformers (Peebles & Xie, 2023) in vision and generative modeling. Such extensions would not only validate the generality of the sequence–graph equivalence across modalities, but also provide a principled framework to analyze and simplify diverse architectures, offering clearer interpretability and guiding the design of more efficient, domain-adaptive models.

## REPRODUCIBILITY STATEMENT

We have taken substantial measures to ensure the reproducibility of our work. The notations, theoretical derivations, and algorithms are thoroughly detailed in Appendix A. Appendix B provides

a complete description of the experimental setup, including training configurations, hyperparameter choices, and algorithmic details.

## STATEMENT ON THE USE OF LARGE LANGUAGE MODELS

We employed a large language model to enhance the manuscript's language, such as improving grammar and phrasing. All research ideas, methods, experiments, analyses, figures/tables, and conclusions were solely developed by the authors, who take full responsibility for the content.

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

# Appendix

## A ADDITIONAL DISCUSSIONS

### A.1 NOTATION OVERVIEW

| Notation | Description |
|---|---|
| $\boldsymbol{G} \in \mathcal{G}$ | Graph with $n$ nodes, node features $\boldsymbol{X}_{n \times d}$, and adjacency matrix $\boldsymbol{A}_{n \times n}$ |
| $\boldsymbol{X}_{n \times d}$ | $n \times d$ node feature matrix; rows are node features $\boldsymbol{x}_i \in \mathbb{R}^d$, $i \in \mathbb{N}_n$ |
| $\boldsymbol{A}_{n \times n}$ | $n \times n$ adjacency matrix encoding edges |
| $\mathbb{N}_n := \{1, \ldots, n\}$ | Set of node indices |
| $\mathbb{N}_S := \{1, \ldots, S\}$ | Set of sequence indices |
| $\boldsymbol{S}_{1:S} \in \mathcal{S}$ | Sequence of length $S$ with features $\boldsymbol{x}_s \in \mathbb{R}^d$, $s \in \mathbb{N}_S$ |
| $\boldsymbol{X}_{S \times d}$ | $S \times d$ sequence feature matrix |
| $\boldsymbol{x} = [x_1, \ldots, x_d]$ | Feature vector as row vector (transpose denotes column form) |
| $\boldsymbol{X}_{(i,:)}$ | $i$-th row of feature matrix $\boldsymbol{X}$ (*i.e.*, $\boldsymbol{e}_i^\top \boldsymbol{X}$) |
| $\boldsymbol{X}_{(:,j)}$ | $j$-th column of feature matrix $\boldsymbol{X}$ (*i.e.*, $\boldsymbol{X} \boldsymbol{e}_j$) |
| $\boldsymbol{e}_i$ | Standard basis vector with 1 in $i$-th position, 0 elsewhere |
| $\boldsymbol{I}$ | Identity matrix |
| $\mathrm{blkdiag}(a_1, \ldots, a_m)$ | Diagonal matrix with entries $a_1, \ldots, a_m$ |
| $\mathrm{blkdiag}(a; m)$ | Diagonal matrix with $m$ identical entries $a$ on the diagonal |

Table 3: Summary of Key Notations.

### A.2 REFORMULATION OF A TRANSFORMER ENCODER

Omitting bias terms, Equation 3 for a Transformer encoder can be expressed as

$$\boldsymbol{X}_{\mathrm{Att}}^{(\ell)} = \mathrm{smx}\left(d^{-1/2} \boldsymbol{X}_{\mathrm{FFN}}^{(\ell-1)} \boldsymbol{W}_Q^{(\ell)} \left(\boldsymbol{X}_{\mathrm{FFN}}^{(\ell-1)} \boldsymbol{W}_K^{(\ell)}\right)^\top\right) \boldsymbol{X}_{\mathrm{FFN}}^{(\ell-1)} \boldsymbol{W}_V^{(\ell)} \tag{11}$$

$$\boldsymbol{X}_{\mathrm{FFN}}^{(\ell)} = \sigma\left(\boldsymbol{X}_{\mathrm{Att}}^{(\ell)} \boldsymbol{W}_{\mathrm{FFN}}^{(\ell,1)}\right) \boldsymbol{W}_{\mathrm{FFN}}^{(\ell,2)} \tag{12}$$

$$\boldsymbol{X}_{\mathrm{Att}}^{(L)} = \boldsymbol{X}_{\mathrm{FFN}}^{(L-1)} \boldsymbol{W}^{(L)}, \tag{13}$$

Substituting Equation 11 into 12 yields:

$$\boldsymbol{X}_{\mathrm{FFN}}^{(\ell)} = \sigma\left(\mathrm{smx}\left(d^{-1/2} \boldsymbol{X}_{\mathrm{FFN}}^{(\ell-1)} \boldsymbol{W}_Q^{(\ell)} \left(\boldsymbol{X}_{\mathrm{FFN}}^{(\ell-1)} \boldsymbol{W}_K^{(\ell)}\right)^\top\right) \boldsymbol{X}_{\mathrm{FFN}}^{(\ell-1)} \boldsymbol{W}_V^{(\ell)} \boldsymbol{W}_{\mathrm{FFN}}^{(\ell,1)}\right) \boldsymbol{W}_{\mathrm{FFN}}^{(\ell,2)} \tag{14}$$

Dropping the subscript for the hidden feature $\boldsymbol{X}_{\mathrm{FFN}}^{(\ell-1)}$ (denoted as $\boldsymbol{X}^{(\ell-1)}$ for brevity) and defining the composite matrix $\boldsymbol{W}_V^{(\ell,1)} := \boldsymbol{W}_V^{(\ell)} \boldsymbol{W}_{\mathrm{FFN}}^{(\ell,1)}$, this simplifies to:

$$\boldsymbol{X}^{(\ell)} = \sigma\left(\mathrm{smx}\left(d^{-1/2} \boldsymbol{X}^{(\ell-1)} \boldsymbol{W}_Q^{(\ell)} \left(\boldsymbol{X}^{(\ell-1)} \boldsymbol{W}_K^{(\ell)}\right)^\top\right) \boldsymbol{X}^{(\ell-1)} \boldsymbol{W}_V^{(\ell,1)}\right) \boldsymbol{W}_{\mathrm{FFN}}^{(\ell,2)}. \tag{15}$$

Thus, the Transformer encoder, derived from Equation 3, is:

$$\boldsymbol{X}^{(\ell)} = \sigma\left(\mathrm{smx}\left(d^{-1/2} \boldsymbol{X}^{(\ell-1)} \boldsymbol{W}_Q^{(\ell)} \left(\boldsymbol{X}^{(\ell-1)} \boldsymbol{W}_K^{(\ell)}\right)^\top\right) \boldsymbol{X}^{(\ell-1)} \boldsymbol{W}_V^{(\ell,1)}\right) \boldsymbol{W}_{\mathrm{FFN}}^{(\ell,2)}$$

$$\boldsymbol{X}^{(L)} = \boldsymbol{X}^{(L-1)} \boldsymbol{W}^{(L)}, \tag{16}$$

### A.3 DERIVATION OF GRADIENTS FOR GRAPH CONVOLUTIONAL NETWORK

For a graph convolutional network (GCN), as defined by Equation 1, a two-layer architecture with multi-hop aggregation can be expressed as:

$$\boldsymbol{X}_{\mathrm{GCN}}^{(2)} = \boldsymbol{A}^{[\kappa_2]} \mathrm{blkdiag}\left(\sigma\left(\boldsymbol{A}^{[\kappa_1]} \mathrm{blkdiag}(\boldsymbol{X}^{(0)}; \kappa_1) \cdot \boldsymbol{W}^{(1)}\right); \kappa_2\right) \cdot \boldsymbol{W}^{(2)}. \tag{17}$$

Using the chain rule, we compute the derivative of $\boldsymbol{X}_{\text{GCN}}^{(2)}$ with respect to the second-layer weights $\boldsymbol{W}^{(2)}$, a vector of dimension $\kappa_2 h_1$, as follows:

$$
\begin{aligned}
\frac{\partial \boldsymbol{X}_{\text{GCN}}^{(2)}}{\partial \boldsymbol{W}^{(2)}} &= \frac{\partial \boldsymbol{A}^{[\kappa_2]}\text{blkdiag}(\boldsymbol{X}^{(1)}; \kappa_2) \cdot \boldsymbol{W}^{(2)}}{\partial \boldsymbol{W}^{(2)}} \\
&= \boldsymbol{A}^{[\kappa_2]}\text{blkdiag}(\boldsymbol{X}^{(1)}; \kappa_2) \\
&= \underbrace{\overbrace{\boldsymbol{A}^{[\kappa_2]}}^{\text{size: } n \times \kappa_2 n} \overbrace{\text{blkdiag}(\sigma(\boldsymbol{A}^{[\kappa_1]}\text{blkdiag}(\boldsymbol{X}^{(0)}; \kappa_1)\boldsymbol{W}^{(1)}); \kappa_2))}^{\text{size: } \kappa_2 n \times \kappa_2 h_1}}_{\text{size: } 1 \times \kappa_2 h_1}.
\end{aligned}
\tag{18}
$$

The derivative with respect to the first-layer weights is more involved. For $i \in \mathbb{N}_{h_1}$

$$
\begin{aligned}
\frac{\partial \boldsymbol{X}_{\text{GCN}}^{(2)}}{\partial \boldsymbol{W}_{(:,i)}^{(1)}} &= \frac{\partial \boldsymbol{A}^{[\kappa_2]}\text{blkdiag}(\boldsymbol{X}^{(1)}\boldsymbol{e}_i\boldsymbol{e}_i^\top; \kappa_2) \cdot \boldsymbol{W}^{(2)}}{\partial \boldsymbol{W}_{(:,i)}^{(1)}} \\
&= \frac{\partial \boldsymbol{A}^{[\kappa_2]}\overbrace{\text{blkdiag}(\boldsymbol{X}^{(1)}\boldsymbol{e}_i; \kappa_2)}^{\text{size: } \kappa_2 n \times \kappa_2} \cdot \overbrace{\text{blkdiag}(\boldsymbol{e}_i^\top; \kappa_2)}^{\text{size: } \kappa_2 \times \kappa_2 h_1} \boldsymbol{W}^{(2)}}{\partial \boldsymbol{W}_{(:,i)}^{(1)}} \\
&= \frac{\partial \boldsymbol{A}^{[\kappa_2]}\overbrace{\text{blkdiag}(\boldsymbol{X}^{(1)}\boldsymbol{e}_i; \kappa_2)}^{\text{size: } \kappa_2 n \times \kappa_2} \cdot \overbrace{\begin{pmatrix} \boldsymbol{W}_{(i-h_1+h_1)}^{(2)} \\ \cdots \\ \boldsymbol{W}_{(i-h_1+\kappa_2 h_1)}^{(2)} \end{pmatrix}}^{\text{size: } \kappa_2 \times 1}}{\partial \boldsymbol{W}_{(:,i)}^{(1)}} \\
&= \frac{\partial \boldsymbol{A}^{[\kappa_2]}\overbrace{\begin{pmatrix} \boldsymbol{X}^{(1)}\boldsymbol{e}_i\boldsymbol{W}_{(i-h_1+h_1)}^{(2)} \\ \cdots \\ \boldsymbol{X}^{(1)}\boldsymbol{e}_i\boldsymbol{W}_{(i-h_1+\kappa_2 h_1)}^{(2)} \end{pmatrix}}^{\text{size: } \kappa_2 n \times 1}}{\partial \boldsymbol{W}_{(:,i)}^{(1)}} \\
&= \boldsymbol{A}^{[\kappa_2]}\begin{pmatrix} \frac{\partial \boldsymbol{X}^{(1)}\boldsymbol{e}_i}{\partial \boldsymbol{W}_{(:,i)}^{(1)}}\boldsymbol{W}_{(i-h_1+h_1)}^{(2)} \\ \cdots \\ \frac{\partial \boldsymbol{X}^{(1)}\boldsymbol{e}_i}{\partial \boldsymbol{W}_{(:,i)}^{(1)}}\boldsymbol{W}_{(i-h_1+\kappa_2 h_1)}^{(2)} \end{pmatrix} \\
&= \underbrace{\overbrace{\boldsymbol{A}^{[\kappa_2]}}^{\text{size: } n \times \kappa_2 n}\overbrace{\begin{pmatrix} \dot{\sigma} \cdot \boldsymbol{A}^{[\kappa_1]}\text{blkdiag}(\boldsymbol{X}^{(0)}; \kappa_1) \cdot \boldsymbol{W}_{(i-h_1+h_1)}^{(2)} \\ \cdots \\ \dot{\sigma} \cdot \boldsymbol{A}^{[\kappa_1]}\text{blkdiag}(\boldsymbol{X}^{(0)}; \kappa_1) \cdot \boldsymbol{W}_{(i-h_1+\kappa_2 h_1)}^{(2)} \end{pmatrix}}^{\text{size: } \kappa_2 n \times \kappa_1 h_0}}_{\text{size: } 1 \times \kappa_1 h_0}},
\end{aligned}
\tag{19}
$$

where the derivative of the activation function is defined as $\dot{\sigma} = \frac{\partial \sigma(\boldsymbol{A}^{[\kappa_1]}\text{blkdiag}(\boldsymbol{X}^{(0)}; \kappa_1)\boldsymbol{W}_{(:,i)}^{(1)})}{\partial \boldsymbol{A}^{[\kappa_1]}\text{blkdiag}(\boldsymbol{X}^{(0)}; \kappa_1)\boldsymbol{W}_{(:,i)}^{(1)}}$.
When using the ReLU activation function, this simplifies to $\dot{\sigma} \cdot \boldsymbol{A}^{[\kappa_1]}\text{blkdiag}(\boldsymbol{X}^{(0)}; \kappa_1) = \sigma\left(\boldsymbol{A}^{[\kappa_1]}\text{blkdiag}(\boldsymbol{X}^{(0)}; \kappa_1)\right)$.

Thus, the gradients with respect to $\boldsymbol{W}^{(2)}$ and $\boldsymbol{W}_{(:,i)}^{(1)}$ (for $i \in \mathbb{N}_{h_1}$) are:

$$
\frac{\partial \boldsymbol{X}_{\text{GCN}}^{(2)}}{\partial \boldsymbol{W}^{(2)}} = \underbrace{\overbrace{\boldsymbol{A}^{[\kappa_2]}}^{n \times \kappa_2 n} \overbrace{\text{blkdiag}(\sigma(\boldsymbol{A}^{[\kappa_1]}\text{blkdiag}(\boldsymbol{X}^{(0)}; \kappa_1)\boldsymbol{W}^{(1)}); \kappa_2)}^{\kappa_2 n \times \kappa_2 h_1}}_{1 \times \kappa_2 h_1},
$$

$$
\frac{\partial \boldsymbol{X}_{\text{GCN}}^{(2)}}{\partial \boldsymbol{W}_{(:,i)}^{(1)}} = \overbrace{\boldsymbol{A}^{[\kappa_2]}}^{n \times \kappa_2 n} \underbrace{\overbrace{\begin{pmatrix} \dot{\sigma} \cdot \boldsymbol{A}^{[\kappa_1]}\text{blkdiag}(\boldsymbol{X}^{(0)}; \kappa_1) \cdot \boldsymbol{W}_{(i-h_1+h_1)}^{(2)} \\ \cdots \\ \dot{\sigma} \cdot \boldsymbol{A}^{[\kappa_1]}\text{blkdiag}(\boldsymbol{X}^{(0)}; \kappa_1) \cdot \boldsymbol{W}_{(i-h_1+\kappa_2 h_1)}^{(2)} \end{pmatrix}}^{\kappa_2 n \times \kappa_1 h_0}}_{1 \times \kappa_1 h_0}. \quad (20)
$$

When restricted to single-hop aggregation (*i.e.*, using only the $\boldsymbol{A}^1$ term), the expressions simplify to:

$$
\frac{\partial \boldsymbol{X}_{\text{GCN}}^{(2)}}{\partial \boldsymbol{W}^{(2)}} = \sigma(\boldsymbol{A}\boldsymbol{X}^{(0)}\boldsymbol{W}^{(1)}), \qquad \frac{\partial \boldsymbol{X}_{\text{GCN}}^{(2)}}{\partial \boldsymbol{W}_{(:,i)}^{(1)}} = \dot{\sigma}\boldsymbol{A}\boldsymbol{X}^{(0)}\boldsymbol{W}_{(i,:)}^{(2)}. \quad (21)
$$

Here, $\boldsymbol{A}$ is a predefined adjacency matrix encoding the graph structure, and $\boldsymbol{W}^{(\ell)}$ governs feature transformation.

### A.4 DERIVATION OF GRADIENTS FOR TRANSFORMER ENCODER

For the Transformer encoder, we simplify Equation 4 by removing the unnecessary linear projection $\boldsymbol{W}_{\text{FFN}}^{(\ell,2)}$ and fixing $L = 2$, resulting in the final layer output:

$$
\boldsymbol{X}_{\text{TF}}^{(2)} = \sigma\left(\text{smx}\left(d^{-1/2}\boldsymbol{X}^{(0)}\boldsymbol{W}_Q^{(1)}\big(\boldsymbol{X}^{(0)}\boldsymbol{W}_K^{(1)}\big)^\top\right)\boldsymbol{X}^{(0)}\boldsymbol{W}_V^{(1,1)}\right) \cdot \boldsymbol{W}^{(2)}, \quad (22)
$$

### A.4.1 GRADIENTS W.R.T. FEATURE TRANSFORMATION PARAMETERS $\boldsymbol{W}^{(2)}$ AND $\boldsymbol{W}_{V(:,i)}^{(1,1)}$

The gradient with respect to $\boldsymbol{W}^{(2)}$ is given by:

$$
\frac{\partial \boldsymbol{X}_{\text{TF}}^{(2)}}{\partial \boldsymbol{W}^{(2)}} = \sigma\left(\text{smx}\left(d^{-1/2}\boldsymbol{X}^{(0)}\boldsymbol{W}_Q^{(1)}\big(\boldsymbol{X}^{(0)}\boldsymbol{W}_K^{(1)}\big)^\top\right)\boldsymbol{X}^{(0)}\boldsymbol{W}_V^{(1,1)}\right) \quad (23)
$$

Similarly, the gradient with respect to $\boldsymbol{W}_{V(:,i)}^{(1,1)}$ ($i \in \mathbb{N}_{h_1}$) is

$$
\begin{aligned}
\frac{\partial \boldsymbol{X}_{\text{TF}}^{(2)}}{\partial \boldsymbol{W}_{V(:,i)}^{(1,1)}} &= \frac{\partial \sigma\left(\text{smx}\left(d^{-1/2}\boldsymbol{X}^{(0)}\boldsymbol{W}_Q^{(1)}\big(\boldsymbol{X}^{(0)}\boldsymbol{W}_K^{(1)}\big)^\top\right)\boldsymbol{X}^{(0)}\boldsymbol{W}_V^{(1,1)}\right) \cdot \boldsymbol{W}^{(2)}}{\partial \boldsymbol{W}_{V(:,i)}^{(1,1)}} \\
&= \dot{\sigma} \cdot \frac{\partial \text{smx}\left(d^{-1/2}\boldsymbol{X}^{(0)}\boldsymbol{W}_Q^{(1)}\big(\boldsymbol{X}^{(0)}\boldsymbol{W}_K^{(1)}\big)^\top\right)\boldsymbol{X}^{(0)}\boldsymbol{W}_V^{(1,1)}}{\partial \boldsymbol{W}_{V(:,i)}^{(1,1)}} \cdot \boldsymbol{W}^{(2)} \\
&= \dot{\sigma} \cdot \text{smx}\left(d^{-1/2}\boldsymbol{X}^{(0)}\boldsymbol{W}_Q^{(1)}\big(\boldsymbol{X}^{(0)}\boldsymbol{W}_K^{(1)}\big)^\top\right)\boldsymbol{X}^{(0)}\boldsymbol{e}_i^\top \cdot \boldsymbol{W}^{(2)} \\
&= \dot{\sigma} \cdot \text{smx}\left(d^{-1/2}\boldsymbol{X}^{(0)}\boldsymbol{W}_Q^{(1)}\big(\boldsymbol{X}^{(0)}\boldsymbol{W}_K^{(1)}\big)^\top\right)\boldsymbol{X}^{(0)} \cdot \boldsymbol{W}_{(i,:)}^{(2)}, \quad (24)
\end{aligned}
$$

where $\dot{\sigma}$ denotes the derivative of the activation function.

### A.4.2 GRADIENTS WITH RESPECT TO QUERY AND KEY WEIGHT MATRICES

Computing the derivative of $\boldsymbol{X}_{\text{TF}}^{(2)}$ with respect to the **query weight matrix** is more complex. For $i \in \mathbb{N}_p$,

$$
\begin{aligned}
\frac{\partial \boldsymbol{X}_{\text{TF}}^{(2)}}{\partial \boldsymbol{W}_{Q(:,i)}^{(1)}} &= \frac{\partial \sigma \left( \text{smx} \left( d^{-1/2} \boldsymbol{X}^{(0)} \boldsymbol{W}_Q^{(1)} \left( \boldsymbol{X}^{(0)} \boldsymbol{W}_K^{(1)} \right)^\top \right) \boldsymbol{X}^{(0)} \boldsymbol{W}_V^{(1,1)} \right) \cdot \boldsymbol{W}^{(2)}}{\partial \boldsymbol{W}_{Q(:,i)}^{(1)}} \\[2mm]
&= \dot{\sigma} \frac{\partial \, \text{smx} \left( d^{-1/2} \boldsymbol{X}^{(0)} \boldsymbol{W}_Q^{(1)} \left( \boldsymbol{X}^{(0)} \boldsymbol{W}_K^{(1)} \right)^\top \right) \boldsymbol{X}^{(0)} \boldsymbol{W}_V^{(1,1)}}{\partial \boldsymbol{W}_{Q(:,i)}^{(1)}} \cdot \boldsymbol{W}^{(2)} \\[2mm]
&= \dot{\sigma} \frac{\partial \, \text{smx} \left( d^{-1/2} \boldsymbol{X}^{(0)} \boldsymbol{W}_Q^{(1)} \boldsymbol{e}_i \boldsymbol{e}_i^\top \left( \boldsymbol{X}^{(0)} \boldsymbol{W}_K^{(1)} \right)^\top \right) \boldsymbol{X}^{(0)} \boldsymbol{W}_V^{(1,1)}}{\partial \boldsymbol{W}_{Q(:,i)}^{(1)}} \cdot \boldsymbol{W}^{(2)} \\[2mm]
&= \begin{bmatrix} \dot{\sigma}_1 \dfrac{\partial \, \text{smx} \left( d^{-1/2} \boldsymbol{X}_{(1,:)}^{(0)} \boldsymbol{W}_{Q(:,i)}^{(1)} \boldsymbol{W}_{K(:,i)}^{(1)}{}^\top \boldsymbol{X}^{(0)\top} \right) \boldsymbol{X}^{(0)} \boldsymbol{W}_V^{(1,1)}}{\partial \boldsymbol{W}_{Q(:,i)}^{(1)}} \cdot \boldsymbol{W}^{(2)} \\ \cdots \\ \dot{\sigma}_S \dfrac{\partial \, \text{smx} \left( d^{-1/2} \boldsymbol{X}_{(S,:)}^{(0)} \boldsymbol{W}_{Q(:,i)}^{(1)} \boldsymbol{W}_{K(:,i)}^{(1)}{}^\top \boldsymbol{X}^{(0)\top} \right) \boldsymbol{X}^{(0)} \boldsymbol{W}_V^{(1,1)}}{\partial \boldsymbol{W}_{Q(:,i)}^{(1)}} \cdot \boldsymbol{W}^{(2)} \end{bmatrix} \\[2mm]
&= \begin{bmatrix} \dot{\sigma}_1 \dfrac{\partial \, \text{smx} \left( d^{-1/2} \boldsymbol{X}_{(1,:)}^{(0)} \boldsymbol{W}_{Q(:,i)}^{(1)} \boldsymbol{W}_{K(:,i)}^{(1)}{}^\top \boldsymbol{X}^{(0)\top} \right)}{\partial \boldsymbol{W}_{Q(:,i)}^{(1)}} \cdot \boldsymbol{X}^{(0)} \boldsymbol{W}_V^{(1,1)} \boldsymbol{W}^{(2)} \\ \cdots \\ \dot{\sigma}_S \dfrac{\partial \, \text{smx} \left( d^{-1/2} \boldsymbol{X}_{(S,:)}^{(0)} \boldsymbol{W}_{Q(:,i)}^{(1)} \boldsymbol{W}_{K(:,i)}^{(1)}{}^\top \boldsymbol{X}^{(0)\top} \right)}{\partial \boldsymbol{W}_{Q(:,i)}^{(1)}} \cdot \boldsymbol{X}^{(0)} \boldsymbol{W}_V^{(1,1)} \boldsymbol{W}^{(2)} \end{bmatrix}
\end{aligned} \tag{25}
$$

To unpack this further, we analyze it row by row. For the $j$-th row ($j \in \mathbb{N}_S$) in Equation 25, the key derivative term is $\dfrac{\partial \, \text{smx} \left( d^{-1/2} \boldsymbol{X}_{(j,:)}^{(0)} \boldsymbol{W}_{Q(:,i)}^{(1)} \boldsymbol{W}_{K(:,i)}^{(1)}{}^\top \boldsymbol{X}^{(0)\top} \right)}{\partial \boldsymbol{W}_{Q(:,i)}^{(1)}}$

$$
\begin{aligned}
&\frac{\partial \, \text{smx} \left( d^{-1/2} \boldsymbol{X}_{(j,:)}^{(0)} \boldsymbol{W}_{Q(:,i)}^{(1)} \boldsymbol{W}_{K(:,i)}^{(1)}{}^\top \boldsymbol{X}^{(0)\top} \right)}{\partial \boldsymbol{W}_{Q(:,i)}^{(1)}} \\[2mm]
&= \frac{\partial \, d^{-1/2} \boldsymbol{X}_{(j,:)}^{(0)} \boldsymbol{W}_{Q(:,i)}^{(1)} \boldsymbol{W}_{K(:,i)}^{(1)}{}^\top \boldsymbol{X}^{(0)\top}}{\partial \boldsymbol{W}_{Q(:,i)}^{(1)}} \frac{\partial \, \text{smx} \left( d^{-1/2} \boldsymbol{X}_{(j,:)}^{(0)} \boldsymbol{W}_{Q(:,i)}^{(1)} \boldsymbol{W}_{K(:,i)}^{(1)}{}^\top \boldsymbol{X}^{(0)\top} \right)}{\partial \, d^{-1/2} \boldsymbol{X}_{(j,:)}^{(0)} \boldsymbol{W}_{Q(:,i)}^{(1)} \boldsymbol{W}_{K(:,i)}^{(1)}{}^\top \boldsymbol{X}^{(0)\top}} \\[2mm]
&= \boldsymbol{X}_{(j,:)}^{(0)} \cdot \frac{1}{\sqrt{d}} \boldsymbol{W}_{K(:,i)}^{(1)}{}^\top \boldsymbol{X}^{(0)\top} \frac{\partial \, \text{smx} \left( d^{-1/2} \boldsymbol{X}_{(j,:)}^{(0)} \boldsymbol{W}_{Q(:,i)}^{(1)} \boldsymbol{W}_{K(:,i)}^{(1)}{}^\top \boldsymbol{X}^{(0)\top} \right)}{\partial \, d^{-1/2} \boldsymbol{X}_{(j,:)}^{(0)} \boldsymbol{W}_{Q(:,i)}^{(1)} \boldsymbol{W}_{K(:,i)}^{(1)}{}^\top \boldsymbol{X}^{(0)\top}} \\[2mm]
&= \boldsymbol{X}_{(j,:)}^{(0)} \cdot \frac{1}{\sqrt{d}} \boldsymbol{W}_{K(:,i)}^{(1)}{}^\top \boldsymbol{X}^{(0)\top} \frac{\partial \left( \dfrac{\exp \left( d^{-1/2} \boldsymbol{X}_{(j,:)}^{(0)} \boldsymbol{W}_{Q(:,i)}^{(1)} \boldsymbol{W}_{K(:,i)}^{(1)}{}^\top \boldsymbol{X}^{(0)\top} \right)}{\boldsymbol{1}^\top \exp \left( d^{-1/2} \boldsymbol{X}_{(j,:)}^{(0)} \boldsymbol{W}_{Q(:,i)}^{(1)} \boldsymbol{W}_{K(:,i)}^{(1)}{}^\top \boldsymbol{X}^{(0)\top} \right)} \right)}{\partial \, d^{-1/2} \boldsymbol{X}_{(j,:)}^{(0)} \boldsymbol{W}_{Q(:,i)}^{(1)} \boldsymbol{W}_{K(:,i)}^{(1)}{}^\top \boldsymbol{X}^{(0)\top}} \\[2mm]
&= \boldsymbol{X}_{(j,:)}^{(0)} \cdot \left( \frac{1}{\sqrt{d}} \boldsymbol{W}_{K(:,i)}^{(1)}{}^\top \boldsymbol{X}^{(0)\top} \text{blkdiag} \left( \text{smx} \left( \boldsymbol{X}_{(j,:)}^{(0)} \boldsymbol{W}_{Q(:,i)}^{(1)} \boldsymbol{W}_{K(:,i)}^{(1)}{}^\top \boldsymbol{X}^{(0)\top} / \sqrt{d} \right) \right) \right. \\[2mm]
&\quad \left. - \frac{1}{\sqrt{d}} \boldsymbol{W}_{K(:,i)}^{(1)}{}^\top \boldsymbol{X}^{(0)\top} \left( \text{smx} \left( \boldsymbol{X}_{(j,:)}^{(0)} \boldsymbol{W}_{Q(:,i)}^{(1)} \boldsymbol{W}_{K(:,i)}^{(1)}{}^\top \boldsymbol{X}^{(0)\top} / \sqrt{d} \right) \right)^\top \text{smx} \left( \boldsymbol{X}_{(j,:)}^{(0)} \boldsymbol{W}_{Q(:,i)}^{(1)} \boldsymbol{W}_{K(:,i)}^{(1)}{}^\top \boldsymbol{X}^{(0)\top} / \sqrt{d} \right) \right)
\end{aligned} \tag{26}
$$

The right-hand side of Equation 26 simplifies to:

$$
\begin{aligned}
&\underbrace{\boldsymbol{X}^{(0)}_{(j,:)}}_{\text{size: }1\times d} \cdot \Bigg( \underbrace{d^{-1/2}}_{1\times 1} \underbrace{\boldsymbol{W}^{(1)}_{K\,(:,i)}}_{1\times d}{}^{\top} \overbrace{\boldsymbol{X}^{(0)}}^{d\times S}{}^{\top} \underbrace{\mathrm{blkdiag}\Big(\mathrm{smx}(\overbrace{\boldsymbol{X}^{(0)}_{(j,:)}}^{1\times d}\overbrace{\boldsymbol{W}^{(1)}_{Q\,(:,i)}}^{d\times 1}\overbrace{\boldsymbol{W}^{(1)}_{K\,(:,i)}}^{1\times d}{}^{\top}\overbrace{\boldsymbol{X}^{(0)}}^{d\times S}{}^{\top}/\sqrt{d})\Big)}_{S\times S} \\
&\qquad -\underbrace{d^{-1/2}}_{1\times 1}\underbrace{\boldsymbol{W}^{(1)}_{K\,(:,i)}}_{1\times d}{}^{\top}\overbrace{\boldsymbol{X}^{(0)}}^{d\times S}{}^{\top}\underbrace{\Big(\mathrm{smx}(\overbrace{\boldsymbol{X}^{(0)}_{(j,:)}}^{1\times d}\overbrace{\boldsymbol{W}^{(1)}_{Q\,(:,i)}}^{d\times 1}\overbrace{\boldsymbol{W}^{(1)}_{K\,(:,i)}}^{1\times d}{}^{\top}\overbrace{\boldsymbol{X}^{(0)}}^{d\times S}{}^{\top}/\sqrt{d})\Big)^{\top}}_{S\times 1}\underbrace{\mathrm{smx}(\overbrace{\boldsymbol{X}^{(0)}_{(j,:)}}^{1\times d}\overbrace{\boldsymbol{W}^{(1)}_{Q\,(:,i)}}^{d\times 1}\overbrace{\boldsymbol{W}^{(1)}_{K\,(:,i)}}^{1\times d}{}^{\top}\overbrace{\boldsymbol{X}^{(0)}}^{d\times S}{}^{\top}/\sqrt{d})}_{1\times S}\Bigg) \\[2mm]
=\ &\underbrace{\boldsymbol{X}^{(0)}_{(j,:)}}_{1\times d} \cdot \Bigg( \underbrace{d^{-1/2}}_{1\times 1}\underbrace{\mathcal{K}^{(1)}_{(:,i)}}_{1\times S}{}^{\top}\underbrace{\mathrm{blkdiag}\Big(\mathrm{smx}(\overbrace{\mathcal{Q}^{(1)}_{(j,i)}}^{1\times 1}\overbrace{\mathcal{K}^{(1)}_{(:,i)}}^{1\times S}{}^{\top}/\sqrt{d})\Big)}_{S\times S} \\
&\qquad -\underbrace{d^{-1/2}}_{1\times 1}\underbrace{\mathcal{K}^{(1)}_{(:,i)}}_{1\times S}{}^{\top}\underbrace{\Big(\mathrm{smx}(\overbrace{\mathcal{Q}^{(1)}_{(j,i)}}^{1\times 1}\overbrace{\mathcal{K}^{(1)}_{(:,i)}}^{1\times S}{}^{\top}/\sqrt{d})\Big)^{\top}}_{S\times 1}\underbrace{\mathrm{smx}(\overbrace{\mathcal{Q}^{(1)}_{(j,i)}}^{1\times 1}\overbrace{\mathcal{K}^{(1)}_{(:,i)}}^{1\times S}{}^{\top}/\sqrt{d})}_{1\times S}\Bigg) \\[2mm]
=\ &d^{-1/2}\underbrace{\boldsymbol{X}^{(0)}_{(j,:)}}_{1\times d}\cdot \underbrace{\Bigg(\overbrace{\mathcal{K}^{(1)}_{(:,i)}}^{1\times S}{}^{\top}\overbrace{\mathrm{blkdiag}\Big(\mathrm{smx}(\xi^{(1)}_{(j,i;:,i)})\Big)}^{S\times S} - \overbrace{\mathcal{K}^{(1)}_{(:,i)}}^{1\times S}{}^{\top}\overbrace{\big(\mathrm{smx}(\xi^{(1)}_{(j,i;:,i)})\big)^{\top}\mathrm{smx}(\xi^{(1)}_{(j,i;:,i)})}^{S\times S}\Bigg)}_{1\times S},
\end{aligned}
$$
$$(27)$$

where $\mathcal{Q}^{(\ell)} := \boldsymbol{X}^{(\ell-1)}\boldsymbol{W}^{(\ell)}_Q$, $\mathcal{K}^{(\ell)} := \boldsymbol{X}^{(\ell-1)}\boldsymbol{W}^{(\ell)}_K$, and $\xi^{(1)}_{(j,i;:,i)} := \mathcal{Q}^{(1)}_{(j,i)}\mathcal{K}^{(1)}_{(:,i)}{}^{\top}/\sqrt{d}$.

Integrating Equations 25, 26, and 27, and dropping the superscripts for $\mathcal{Q}$ and $\mathcal{K}$ for brevity, yields:

$$
\begin{aligned}
&\frac{\partial \boldsymbol{X}^{(2)}_{\mathrm{TF}}}{\partial \boldsymbol{W}^{(1)}_{Q\,(:,i)}} \\
=\ &\begin{bmatrix}
\dot{\sigma}_1/\sqrt{d}\,\underbrace{\boldsymbol{X}^{(0)}_{(1,:)}}_{1\times d}\cdot \underbrace{\Big(\overbrace{\mathcal{K}^{(1)}_{(:,i)}}^{1\times S}{}^{\top}\overbrace{\mathrm{blkdiag}\big(\mathrm{smx}(\xi^{(1)}_{(1,i;:,i)})\big)}^{S\times S}\overbrace{\boldsymbol{X}^{(0)}}^{S\times d}\overbrace{\boldsymbol{W}^{(1,1)}_V}^{d\times h_1}\overbrace{\boldsymbol{W}^{(2)}}^{h_1\times 1} - \overbrace{\mathcal{K}^{(1)}_{(:,i)}}^{1\times S}{}^{\top}\overbrace{\big(\mathrm{smx}(\xi^{(1)}_{(1,i;:,i)})\big)^{\top}\mathrm{smx}(\xi^{(1)}_{(1,i;:,i)})}^{S\times S}\overbrace{\boldsymbol{X}^{(0)}}^{S\times d}\overbrace{\boldsymbol{W}^{(1,1)}_V}^{d\times h_1}\overbrace{\boldsymbol{W}^{(2)}}^{h_1\times 1}\Big)}_{1\times 1} \\
\cdots \\
\dot{\sigma}_S/\sqrt{d}\,\underbrace{\boldsymbol{X}^{(0)}_{(S,:)}}_{1\times d}\cdot \underbrace{\Big(\overbrace{\mathcal{K}^{(1)}_{(:,i)}}^{1\times S}{}^{\top}\overbrace{\mathrm{blkdiag}\big(\mathrm{smx}(\xi^{(1)}_{(S,i;:,i)})\big)}^{S\times S}\overbrace{\boldsymbol{X}^{(0)}}^{S\times d}\overbrace{\boldsymbol{W}^{(1,1)}_V}^{d\times h_1}\overbrace{\boldsymbol{W}^{(2)}}^{h_1\times 1} - \overbrace{\mathcal{K}^{(1)}_{(:,i)}}^{1\times S}{}^{\top}\overbrace{\big(\mathrm{smx}(\xi^{(1)}_{(S,i;:,i)})\big)^{\top}\mathrm{smx}(\xi^{(1)}_{(S,i;:,i)})}^{S\times S}\overbrace{\boldsymbol{X}^{(0)}}^{S\times d}\overbrace{\boldsymbol{W}^{(1,1)}_V}^{d\times h_1}\overbrace{\boldsymbol{W}^{(2)}}^{h_1\times 1}\Big)}_{1\times 1}
\end{bmatrix}_{S\times d} \\[2mm]
=\ &\Big[\dot{\sigma}_j/\sqrt{d}\,\boldsymbol{X}^{(0)}_{(j,:)}\cdot\Big(\mathcal{K}_{(:,i)}{}^{\top}\mathrm{blkdiag}\big(\mathrm{smx}(\xi^{(1)}_{(j,i;:,i)})\big) - \mathcal{K}_{(:,i)}{}^{\top}\big(\mathrm{smx}(\xi^{(1)}_{(j,i;:,i)})\big)^{\top}\mathrm{smx}(\xi^{(1)}_{(j,i;:,i)})\Big)\cdot\boldsymbol{X}^{(0)}\boldsymbol{W}^{(1,1)}_V\boldsymbol{W}^{(2)}\Big]_{S\times d} \\[2mm]
=\ &\mathrm{blkdiag}\Big(\dot{\sigma}_1/\sqrt{d}\Big(\mathcal{K}_{(:,i)}{}^{\top}\mathrm{blkdiag}\big(\mathrm{smx}(\xi^{(1)}_{(1,i;:,i)})\big) - \mathcal{K}_{(:,i)}{}^{\top}\big(\mathrm{smx}(\xi^{(1)}_{(1,i;:,i)})\big)^{\top}\mathrm{smx}(\xi^{(1)}_{(1,i;:,i)})\Big)\boldsymbol{X}^{(0)}\boldsymbol{W}^{(1,1)}_V\boldsymbol{W}^{(2)}, \\
&\qquad \cdots ,\dot{\sigma}_S/\sqrt{d}\Big(\mathcal{K}_{(:,i)}{}^{\top}\mathrm{blkdiag}\big(\mathrm{smx}(\xi^{(1)}_{(S,i;:,i)})\big) - \mathcal{K}_{(:,i)}{}^{\top}\big(\mathrm{smx}(\xi^{(1)}_{(S,i;:,i)})\big)^{\top}\mathrm{smx}(\xi^{(1)}_{(S,i;:,i)})\Big)\boldsymbol{X}^{(0)}\boldsymbol{W}^{(1,1)}_V\boldsymbol{W}^{(2)}\Big)\boldsymbol{X}^{(0)}.
\end{aligned}
$$
$$(28)$$

The derivative with respect to the **key weight matrix** follows a parallel approach. For $i \in \mathbb{N}_p$,

$$
\begin{aligned}
\frac{\partial \boldsymbol{X}_{\mathrm{TF}}^{(2)}}{\partial \boldsymbol{W}_{K\,(:,i)}^{(1)}} &= \frac{\partial\,\sigma\Big(\mathrm{smx}\Big(d^{-1/2}\boldsymbol{X}^{(0)}\boldsymbol{W}_Q^{(1)}\big(\boldsymbol{X}^{(0)}\boldsymbol{W}_K^{(1)}\big)^{\top}\Big)\boldsymbol{X}^{(0)}\boldsymbol{W}_V^{(1,1)}\Big) \cdot \boldsymbol{W}^{(2)}}{\partial \boldsymbol{W}_{K\,(:,i)}^{(1)}} \\[2mm]
&= \dot{\sigma}\,\frac{\partial\,\mathrm{smx}\Big(d^{-1/2}\boldsymbol{X}^{(0)}\boldsymbol{W}_Q^{(1)}\big(\boldsymbol{X}^{(0)}\boldsymbol{W}_K^{(1)}\big)^{\top}\Big)\boldsymbol{X}^{(0)}\boldsymbol{W}_V^{(1,1)}}{\partial \boldsymbol{W}_{K\,(:,i)}^{(1)}} \cdot \boldsymbol{W}^{(2)} \\[2mm]
&= \dot{\sigma}\,\frac{\partial\,\mathrm{smx}\Big(d^{-1/2}\boldsymbol{X}^{(0)}\boldsymbol{W}_Q^{(1)}\big(\boldsymbol{X}^{(0)}\boldsymbol{W}_K^{(1)}\boldsymbol{e}_i\boldsymbol{e}_i^{\top}\big)^{\top}\Big)\boldsymbol{X}^{(0)}\boldsymbol{W}_V^{(1,1)}}{\partial \boldsymbol{W}_{K\,(:,i)}^{(1)}} \cdot \boldsymbol{W}^{(2)} \\[2mm]
&= \begin{bmatrix} \dot{\sigma}_1\,\dfrac{\partial\,\mathrm{smx}\Big(d^{-1/2}\boldsymbol{X}_{(1,:)}^{(0)}\boldsymbol{W}_{Q\,(:,i)}^{(1)}\boldsymbol{W}_{K\,(:,i)}^{(1)}{}^{\top}\boldsymbol{X}^{(0)\top}\Big)\boldsymbol{X}^{(0)}\boldsymbol{W}_V^{(1,1)}}{\partial \boldsymbol{W}_{K\,(:,i)}^{(1)}} \cdot \boldsymbol{W}^{(2)} \\ \cdots \\ \dot{\sigma}_S\,\dfrac{\partial\,\mathrm{smx}\Big(d^{-1/2}\boldsymbol{X}_{(S,:)}^{(0)}\boldsymbol{W}_{Q\,(:,i)}^{(1)}\boldsymbol{W}_{K\,(:,i)}^{(1)}{}^{\top}\boldsymbol{X}^{(0)\top}\Big)\boldsymbol{X}^{(0)}\boldsymbol{W}_V^{(1,1)}}{\partial \boldsymbol{W}_{K\,(:,i)}^{(1)}} \cdot \boldsymbol{W}^{(2)} \end{bmatrix} \\[2mm]
&= \begin{bmatrix} \dot{\sigma}_1\,\dfrac{\partial\,\mathrm{smx}\Big(d^{-1/2}\boldsymbol{X}_{(1,:)}^{(0)}\boldsymbol{W}_{Q\,(:,i)}^{(1)}\boldsymbol{W}_{K\,(:,i)}^{(1)}{}^{\top}\boldsymbol{X}^{(0)\top}\Big)}{\partial \boldsymbol{W}_{K\,(:,i)}^{(1)}} \cdot \boldsymbol{X}^{(0)}\boldsymbol{W}_V^{(1,1)}\boldsymbol{W}^{(2)} \\ \cdots \\ \dot{\sigma}_S\,\dfrac{\partial\,\mathrm{smx}\Big(d^{-1/2}\boldsymbol{X}_{(S,:)}^{(0)}\boldsymbol{W}_{Q\,(:,i)}^{(1)}\boldsymbol{W}_{K\,(:,i)}^{(1)}{}^{\top}\boldsymbol{X}^{(0)\top}\Big)}{\partial \boldsymbol{W}_{K\,(:,i)}^{(1)}} \cdot \boldsymbol{X}^{(0)}\boldsymbol{W}_V^{(1,1)}\boldsymbol{W}^{(2)} \end{bmatrix}
\end{aligned}
\tag{29}
$$

Again, examining row by row for the $j$-th row ($j \in \mathbb{N}_S$) in Equation 29, the focal derivative is $\frac{\partial\,\mathrm{smx}\Big(d^{-1/2}\boldsymbol{X}_{(j,:)}^{(0)}\boldsymbol{W}_{Q\,(:,i)}^{(1)}\boldsymbol{W}_{K\,(:,i)}^{(1)}{}^{\top}\boldsymbol{X}^{(0)\top}\Big)}{\partial \boldsymbol{W}_{K\,(:,i)}^{(1)}}$:

$$
\begin{aligned}
&\frac{\partial\,\mathrm{smx}\Big(d^{-1/2}\boldsymbol{X}_{(j,:)}^{(0)}\boldsymbol{W}_{Q\,(:,i)}^{(1)}\boldsymbol{W}_{K\,(:,i)}^{(1)}{}^{\top}\boldsymbol{X}^{(0)\top}\Big)}{\partial \boldsymbol{W}_{K\,(:,i)}^{(1)}} \\[2mm]
&= \frac{\partial\,d^{-1/2}\boldsymbol{X}_{(j,:)}^{(0)}\boldsymbol{W}_{Q\,(:,i)}^{(1)}\boldsymbol{W}_{K\,(:,i)}^{(1)}{}^{\top}\boldsymbol{X}^{(0)\top}}{\partial \boldsymbol{W}_{K\,(:,i)}^{(1)}}\,\frac{\partial\,\mathrm{smx}\Big(d^{-1/2}\boldsymbol{X}_{(j,:)}^{(0)}\boldsymbol{W}_{Q\,(:,i)}^{(1)}\boldsymbol{W}_{K\,(:,i)}^{(1)}{}^{\top}\boldsymbol{X}^{(0)\top}\Big)}{\partial\,d^{-1/2}\boldsymbol{X}_{(j,:)}^{(0)}\boldsymbol{W}_{Q\,(:,i)}^{(1)}\boldsymbol{W}_{K\,(:,i)}^{(1)}{}^{\top}\boldsymbol{X}^{(0)\top}} \\[2mm]
&= \boldsymbol{X}_{(j,:)}^{(0)} \cdot \frac{1}{\sqrt{d}}\boldsymbol{W}_{Q\,(:,i)}^{(1)}{}^{\top}\boldsymbol{X}^{(0)\top}\,\frac{\partial\,\mathrm{smx}\Big(d^{-1/2}\boldsymbol{X}_{(j,:)}^{(0)}\boldsymbol{W}_{Q\,(:,i)}^{(1)}\boldsymbol{W}_{K\,(:,i)}^{(1)}{}^{\top}\boldsymbol{X}^{(0)\top}\Big)}{\partial\,d^{-1/2}\boldsymbol{X}_{(j,:)}^{(0)}\boldsymbol{W}_{Q\,(:,i)}^{(1)}\boldsymbol{W}_{K\,(:,i)}^{(1)}{}^{\top}\boldsymbol{X}^{(0)\top}} \\[2mm]
&= \boldsymbol{X}_{(j,:)}^{(0)} \cdot \frac{1}{\sqrt{d}}\boldsymbol{W}_{Q\,(:,i)}^{(1)}{}^{\top}\boldsymbol{X}^{(0)\top}\,\frac{\partial\left(\dfrac{\exp\Big(d^{-1/2}\boldsymbol{X}_{(j,:)}^{(0)}\boldsymbol{W}_{Q\,(:,i)}^{(1)}\boldsymbol{W}_{K\,(:,i)}^{(1)}{}^{\top}\boldsymbol{X}^{(0)\top}\Big)}{\boldsymbol{1}^{\top}\exp\Big(d^{-1/2}\boldsymbol{X}_{(j,:)}^{(0)}\boldsymbol{W}_{Q\,(:,i)}^{(1)}\boldsymbol{W}_{K\,(:,i)}^{(1)}{}^{\top}\boldsymbol{X}^{(0)\top}\Big)}\right)}{\partial\,d^{-1/2}\boldsymbol{X}_{(j,:)}^{(0)}\boldsymbol{W}_{Q\,(:,i)}^{(1)}\boldsymbol{W}_{K\,(:,i)}^{(1)}{}^{\top}\boldsymbol{X}^{(0)\top}} \\[2mm]
&= \boldsymbol{X}_{(j,:)}^{(0)} \cdot \Big(\frac{1}{\sqrt{d}}\boldsymbol{W}_{Q\,(:,i)}^{(1)}{}^{\top}\boldsymbol{X}^{(0)\top}\,\mathrm{blkdiag}\big(\mathrm{smx}(\boldsymbol{X}_{(j,:)}^{(0)}\boldsymbol{W}_{Q\,(:,i)}^{(1)}\boldsymbol{W}_{K\,(:,i)}^{(1)}{}^{\top}\boldsymbol{X}^{(0)\top}/\sqrt{d}) \big) \\[2mm]
&\quad -\frac{1}{\sqrt{d}}\boldsymbol{W}_{Q\,(:,i)}^{(1)}{}^{\top}\boldsymbol{X}^{(0)\top}\big(\mathrm{smx}(\boldsymbol{X}_{(j,:)}^{(0)}\boldsymbol{W}_{Q\,(:,i)}^{(1)}\boldsymbol{W}_{K\,(:,i)}^{(1)}{}^{\top}\boldsymbol{X}^{(0)\top}/\sqrt{d})\big)^{\top}\mathrm{smx}(\boldsymbol{X}_{(j,:)}^{(0)}\boldsymbol{W}_{Q\,(:,i)}^{(1)}\boldsymbol{W}_{K\,(:,i)}^{(1)}{}^{\top}\boldsymbol{X}^{(0)\top}/\sqrt{d})\Big)
\end{aligned}
\tag{30}
$$

Simplifying the right-hand side of Equation 30 gives:

$$
\begin{aligned}
&\underbrace{\boldsymbol{X}^{(0)}_{(j,:)}}_{\text{size: } 1\times d} \cdot \Bigg( \underbrace{d^{-1/2}}_{1\times 1} \underbrace{\boldsymbol{W}^{(1)\top}_{Q\,(:,i)}}_{1\times d} \overbrace{\boldsymbol{X}^{(0)\top}}^{d\times S} \underbrace{\mathrm{blkdiag}\Big(\mathrm{smx}(\overbrace{\boldsymbol{X}^{(0)}_{(j,:)}}^{1\times d} \overbrace{\boldsymbol{W}^{(1)}_{Q\,(:,i)}}^{d\times 1} \overbrace{\boldsymbol{W}^{(1)\top}_{K\,(:,i)}}^{1\times d} \overbrace{\boldsymbol{X}^{(0)\top}}^{d\times S}/\sqrt{d})\Big)}_{S\times S} \\
&\quad - \underbrace{d^{-1/2}}_{1\times 1} \underbrace{\boldsymbol{W}^{(1)\top}_{Q\,(:,i)}}_{1\times d} \overbrace{\boldsymbol{X}^{(0)\top}}^{d\times S} \underbrace{\Big(\mathrm{smx}(\boldsymbol{X}^{(0)}_{(j,:)}\boldsymbol{W}^{(1)}_{Q\,(:,i)}\boldsymbol{W}^{(1)\top}_{K\,(:,i)}\boldsymbol{X}^{(0)\top}/\sqrt{d})\Big)^{\top}}_{S\times 1} \underbrace{\mathrm{smx}(\boldsymbol{X}^{(0)}_{(j,:)}\boldsymbol{W}^{(1)}_{Q\,(:,i)}\boldsymbol{W}^{(1)\top}_{K\,(:,i)}\boldsymbol{X}^{(0)\top}/\sqrt{d})}_{1\times S} \Bigg) \\[4pt]
=\ &\underbrace{\boldsymbol{X}^{(0)}_{(j,:)}}_{1\times d} \cdot \Bigg( \underbrace{d^{-1/2}}_{1\times 1} \underbrace{\mathcal{Q}^{(1)\top}_{(:,i)}}_{1\times S} \underbrace{\mathrm{blkdiag}\Big(\mathrm{smx}(\overbrace{\mathcal{Q}^{(1)}_{(j,i)}}^{1\times 1}\overbrace{\mathcal{K}^{(1)\top}_{(:,i)}}^{1\times S}/\sqrt{d})\Big)}_{S\times S} \\
&\quad - \underbrace{d^{-1/2}}_{1\times 1} \underbrace{\mathcal{Q}^{(1)\top}_{(:,i)}}_{1\times S} \underbrace{\Big(\mathrm{smx}(\overbrace{\mathcal{Q}^{(1)}_{(j,i)}}^{1\times 1}\overbrace{\mathcal{K}^{(1)\top}_{(:,i)}}^{1\times S}/\sqrt{d})\Big)^{\top}}_{S\times 1} \underbrace{\mathrm{smx}(\overbrace{\mathcal{Q}^{(1)}_{(j,i)}}^{1\times 1}\overbrace{\mathcal{K}^{(1)\top}_{(:,i)}}^{1\times S}/\sqrt{d})}_{1\times S} \Bigg) \\[4pt]
=\ &d^{-1/2}\underbrace{\boldsymbol{X}^{(0)}_{(j,:)}}_{1\times d} \cdot \underbrace{\Bigg( \overbrace{\mathcal{Q}^{(1)\top}_{(:,i)}}^{1\times S} \overbrace{\mathrm{blkdiag}\big(\mathrm{smx}(\xi^{(1)}_{(j,i;:,i)})\big)}^{S\times S} - \overbrace{\mathcal{Q}^{(1)\top}_{(:,i)}}^{1\times S} \overbrace{\big(\mathrm{smx}(\xi^{(1)}_{(j,i;:,i)})\big)^{\top}\mathrm{smx}(\xi^{(1)}_{(j,i;:,i)})}^{S\times S} \Bigg)}_{1\times S},
\end{aligned}
\tag{31}
$$

Combining Equations 29, 30, and 31 results in:

$$
\begin{aligned}
&\frac{\partial \boldsymbol{X}^{(2)}_{\mathrm{TF}}}{\partial \boldsymbol{W}^{(1)}_{K\,(:,i)}} \\
=\ &\begin{bmatrix} \dot{\sigma}_1/\sqrt{d}\,\underbrace{\boldsymbol{X}^{(0)}_{(1,:)}}_{1\times d}\cdot\underbrace{\Big( \overbrace{\mathcal{Q}^{(1)\top}_{(:,i)}}^{1\times S}\overbrace{\mathrm{blkdiag}\big(\mathrm{smx}(\xi^{(1)}_{(1,i;:,i)})\big)}^{S\times S}\overbrace{\boldsymbol{X}^{(0)}}^{S\times d}\overbrace{\boldsymbol{W}^{(1,1)}_{V}}^{d\times h_1}\overbrace{\boldsymbol{W}^{(2)}}^{h_1\times 1} - \overbrace{\mathcal{Q}^{(1)\top}_{(:,i)}}^{1\times S}\overbrace{\big(\mathrm{smx}(\xi^{(1)}_{(1,i;:,i)})\big)^{\top}\mathrm{smx}(\xi^{(1)}_{(1,i;:,i)})}^{S\times S}\overbrace{\boldsymbol{X}^{(0)}}^{S\times d}\overbrace{\boldsymbol{W}^{(1,1)}_{V}}^{d\times h_1}\overbrace{\boldsymbol{W}^{(2)}}^{h_1\times 1}\Big)}_{1\times 1} \\ \cdots \\ \dot{\sigma}_S/\sqrt{d}\,\underbrace{\boldsymbol{X}^{(0)}_{(S,:)}}_{1\times d}\cdot\underbrace{\Big(\overbrace{\mathcal{Q}^{(1)\top}_{(:,i)}}^{1\times S}\overbrace{\mathrm{blkdiag}\big(\mathrm{smx}(\xi^{(1)}_{(S,i;:,i)})\big)}^{S\times S}\overbrace{\boldsymbol{X}^{(0)}}^{S\times d}\overbrace{\boldsymbol{W}^{(1,1)}_{V}}^{d\times h_1}\overbrace{\boldsymbol{W}^{(2)}}^{h_1\times 1} - \overbrace{\mathcal{Q}^{(1)\top}_{(:,i)}}^{1\times S}\overbrace{\big(\mathrm{smx}(\xi^{(1)}_{(S,i;:,i)})\big)^{\top}\mathrm{smx}(\xi^{(1)}_{(S,i;:,i)})}^{S\times S}\overbrace{\boldsymbol{X}^{(0)}}^{S\times d}\overbrace{\boldsymbol{W}^{(1,1)}_{V}}^{d\times h_1}\overbrace{\boldsymbol{W}^{(2)}}^{h_1\times 1}\Big)}_{1\times 1} \end{bmatrix}_{S\times d} \\[4pt]
=\ &\begin{bmatrix} \dot{\sigma}_j/\sqrt{d}\,\boldsymbol{X}^{(0)}_{(j,:)}\cdot\Big(\mathcal{Q}^{(1)\top}_{(:,i)}\mathrm{blkdiag}\big(\mathrm{smx}(\xi^{(1)}_{(j,i;:,i)})\big)-\mathcal{Q}^{(1)\top}_{(:,i)}\big(\mathrm{smx}(\xi^{(1)}_{(j,i;:,i)})\big)^{\top}\mathrm{smx}(\xi^{(1)}_{(j,i;:,i)})\Big)\cdot\boldsymbol{X}^{(0)}\boldsymbol{W}^{(1,1)}_{V}\boldsymbol{W}^{(2)} \end{bmatrix}_{S\times d} \\[4pt]
=\ &\mathrm{blkdiag}\bigg( \dot{\sigma}_1/\sqrt{d}\,\Big(\mathcal{Q}^{(1)\top}_{(:,i)}\mathrm{blkdiag}\big(\mathrm{smx}(\xi^{(1)}_{(1,i;:,i)})\big)-\mathcal{Q}^{(1)\top}_{(:,i)}\big(\mathrm{smx}(\xi^{(1)}_{(1,i;:,i)})\big)^{\top}\mathrm{smx}(\xi^{(1)}_{(1,i;:,i)})\Big)\boldsymbol{X}^{(0)}\boldsymbol{W}^{(1,1)}_{V}\boldsymbol{W}^{(2)}, \\
&\qquad \cdots, \dot{\sigma}_S/\sqrt{d}\,\Big(\mathcal{Q}^{(1)\top}_{(:,i)}\mathrm{blkdiag}\big(\mathrm{smx}(\xi^{(1)}_{(S,i;:,i)})\big)-\mathcal{Q}^{(1)\top}_{(:,i)}\big(\mathrm{smx}(\xi^{(1)}_{(S,i;:,i)})\big)^{\top}\mathrm{smx}(\xi^{(1)}_{(S,i;:,i)})\Big)\boldsymbol{X}^{(0)}\boldsymbol{W}^{(1,1)}_{V}\boldsymbol{W}^{(2)}\bigg)\boldsymbol{X}^{(0)}.
\end{aligned}
\tag{32}
$$

These derivations naturally extend to scenarios where each component of the output $\boldsymbol{X}^{(L)}_{\mathrm{TF}}$ is a vector, through a multi-dimensional framework (Micchelli & Pontil, 2005; Zhang et al., 2023). For multi-head attention, the process can be parallelized by replicating the derivations across the number of heads.

## A.5 FIGHTER FORWARD PASS (WITH OPTIONAL ENHANCEMENTS)

---

**Algorithm 1** Fighter Forward Pass (with Optional Enhancements)

---

**Require:** Input features $\boldsymbol{X}_{\text{FT}}^{(0)} \in \mathbb{R}^{S \times d}$, temporary variable $\boldsymbol{A}$, number of layers $L$, hop parameters $\{\kappa_\ell\}_{\ell=1}^L$, weights $\{\boldsymbol{W}_Q^{(\ell)}, \boldsymbol{W}_K^{(\ell)}, \boldsymbol{W}^{(\ell)}\}_{\ell=1}^L$

**Ensure:** Output $\boldsymbol{X}_{\text{FT}}^{(L)}$

1: $\boldsymbol{X} \leftarrow \boldsymbol{X}_{\text{FT}}^{(0)}$ ▷ Initialize with input features
2: **for** $\ell = 1$ to $L - 1$ **do**
3:     *(Optional)*: $\tilde{\boldsymbol{X}} \leftarrow \text{LayerNorm}(\boldsymbol{X})$ ▷ Apply layer normalization
4:     $\boldsymbol{Q} \leftarrow \tilde{\boldsymbol{X}} \boldsymbol{W}_Q^{(\ell)}$ ▷ Queries; use $\boldsymbol{X}$ if no norm
5:     $\boldsymbol{K} \leftarrow \tilde{\boldsymbol{X}} \boldsymbol{W}_K^{(\ell)}$ ▷ Keys; use $\boldsymbol{X}$ if no norm
6:     *(Optional)*: Compute $\boldsymbol{Q}, \boldsymbol{K}$ for each head, concatenate/project outputs
7:     $\boldsymbol{A} \leftarrow \text{smx}(d^{-1/2} \boldsymbol{Q} \boldsymbol{K}^\top)$ ▷ Compute attention distribution matrices
8:     $\tilde{\boldsymbol{A}} \leftarrow \boldsymbol{A}^{[\kappa_\ell]}$ ▷ Multi-hop attention: raise to powers up to $\kappa_\ell - 1$
9:     $\tilde{\boldsymbol{X}} \leftarrow \text{blkdiag}(\tilde{\boldsymbol{X}}; \kappa_\ell)$ ▷ Block-diagonal replication; use $\boldsymbol{X}$ if no norm
10:    $\tilde{\boldsymbol{X}} \leftarrow \tilde{\boldsymbol{A}} \tilde{\boldsymbol{X}} \boldsymbol{W}^{(\ell)}$ ▷ Aggregate and project with streamlined weight
11:    $\boldsymbol{X} \leftarrow \sigma(\tilde{\boldsymbol{X}})$ ▷ Apply activation (*e.g.*, ReLU)
12:    *(Optional)*: $\boldsymbol{X} \leftarrow \boldsymbol{X} + \tilde{\boldsymbol{X}}$ ▷ Add residual; use $\boldsymbol{X}$ if no norm
13: **end for**
14: *(Optional)*: $\tilde{\boldsymbol{X}} \leftarrow \text{LayerNorm}(\boldsymbol{X})$ ▷ Final layer normalization
15: $\boldsymbol{Q} \leftarrow \tilde{\boldsymbol{X}} \boldsymbol{W}_Q^{(L)}$ ▷ Final queries; use $\boldsymbol{X}$ if no norm
16: $\boldsymbol{K} \leftarrow \tilde{\boldsymbol{X}} \boldsymbol{W}_K^{(L)}$ ▷ Final keys; use $\boldsymbol{X}$ if no norm
17: *(Optional)*: Compute $\boldsymbol{Q}, \boldsymbol{K}$ for each head, concatenate/project outputs
18: $\boldsymbol{A} \leftarrow \text{smx}(d^{-1/2} \boldsymbol{Q} \boldsymbol{K}^\top)$ ▷ Final attention distribution matrices
19: $\tilde{\boldsymbol{A}} \leftarrow \boldsymbol{A}^{[\kappa_\ell]}$ ▷ Final multi-hop attention
20: $\tilde{\boldsymbol{X}} \leftarrow \text{blkdiag}(\tilde{\boldsymbol{X}}; \kappa_\ell)$ ▷ Final block-diagonal replication; use $\boldsymbol{X}$ if no norm
21: $\boldsymbol{X}_{\text{FT}}^{(L)} \leftarrow \tilde{\boldsymbol{A}}^{[\kappa_L]} \tilde{\boldsymbol{X}} \boldsymbol{W}^{(L)}$ ▷ Final output
22: *(Optional)*: $\boldsymbol{X}_{\text{FT}}^{(L)} \leftarrow \boldsymbol{X}_{\text{FT}}^{(L)} + \tilde{\boldsymbol{X}}$ ▷ Add final residual; use $\boldsymbol{X}$ if no norm **return** $\boldsymbol{X}_{\text{FT}}^{(L)}$

---

# B  Experiment Details

This section outlines the experiment details, covering the experimental setup, supplementary results, and a brief sensitivity analysis on benchmark datasets.

## B.1  Experimental Setup

**Device Setup.** We mainly conduct experiments using NVIDIA Geforce RTX 3090 (24G).

**Dataset Splitting.** The train / val / test split configurations for the benchmark datasets are provided in Table 4. We adopt the Electricity, Weather and ETTh1, ETTm1 datasets for time series forecasting. All time series inputs are standardized using a StandardScaler(Pedregosa et al., 2011).

| Dataset | train | validation | test |
|---|---|---|---|
| Electricity | 70% | 10% | 20% |
| Weather | 70% | 10% | 20% |
| ETTh1 | 60% | 20% | 20% |
| ETTm1 | 60% | 20% | 20% |
| AG News | 90% | 0% | 10% |

Table 4: Dataset splitting for the benchmark datasets.

**Hyperparameter Settings.** The key hyperparameter settings are listed in Table 5. The prediction lengths are all given {96, 192, 336, 720}.

| Dataset | n-blocks | $\kappa-$list | batch-size | input-len | epochs |
|---|---|---|---|---|---|
| Electricity | 1 | [3] | 128 | 96 | 25 |
| Weather | 1 | [3] | 128 | 96 | 25 |
| ETTh1 | 1 | [3] | 32 | 96 | 25 |
| ETTm1 | 1 | [3] | 32 | 96 | 25 |
| AG News | 1 | [3] | 32 | - | 25 |

Table 5: Key hyperparameter settings on the benchmark datasets for Fighter.

We also adopt the `ReduceLROnPlateau` (Hinton et al., 2012) as the learning rate scheduler, and the early stopping for the ETTh1 and ETTm1 datasets.

## B.2  Sensitivity Analysis

We present attention heatmaps for the first 64 positions at four training moments (25, 50, 75, and 100 rounds) and four kappa settings, *i.e.*, $\kappa \in \{1, 2, 3, 4\}$. As $\kappa$ increases, a position is more likely to activate the most relevant position, effectively expanding the receptive field. By adjusting $\kappa$ in time series and text classification tasks, Fighter gradually forms a collaborative modeling of global and local information, improving its representation capabilities.

The attention heatmaps in Table 8 provide a detailed visualization of Fighter and the Transformer model, illustrating the impact of higher-order dynamic attention mechanism and the connection to graph-based feature aggregation. This paper reinterprets the Transformer encoder through the lens of graph convolution, where the attention distribution matrix acts as a dynamic adjacency matrix. In this context, Fighter extends the Transformer by incorporating higher-order compositions of the attention distribution matrix, controlled by the parameter $\kappa$, which governs the power to which the attention distributions are raised. This flexible mechanism enables Fighter to perform multi-hop graph aggregation, capturing temporal dependencies across multiple scales. The heatmaps vividly demonstrate how $\kappa$ transforms the Fighter's attention patterns, evolving from localized dependencies to capturing broader, global dependencies.

It can be observed that the Transformer's attention heatmap is constrained to local dependencies, as reflected in its strong diagonal focus. While this is effective for short-range temporal modeling, it limits the Transformer's capacity to capture long-range dependencies and multi-scale relationships.

Fighter, on the other hand, introduces higher-order attention distribution matrix to aggregate information across multiple hops in the time series. At $\kappa = 2$, Fighter's attention heatmap resembles the Transformer's, emphasizing local interactions. However, as $\kappa$ increases, the Fighter's attention becomes more distributed and structured, capturing long-range dependencies and revealing richer temporal patterns. This behavior aligns with the theoretical interpretation of Fighter as performing higher-order graph convolution, where increasing $\kappa$ corresponds to aggregating information from wider neighborhoods in the graph.

The incorporation of higher-order attention mechanisms in Fighter provides several key advantages. By exponentiating the attention distribution matrix, Fighter flexibly emphasizes stronger relationships and suppresses weaker ones, effectively controlling the breadth of the graph aggregation process. This mechanism not only improves the model's ability to capture complex temporal dependencies but also provides an interpretable representation of these dependencies as graph edges. The parameter $\kappa$ acts as a tunable knob, balancing local and global attention patterns, thereby enabling Fighter to adapt to the specific demands of the task. This flexibility, combined with the removal of redundant linear projections, results in a streamlined architecture that is both computationally efficient and interpretable.

Fighter bridges the gap between Transformers and GCNs, offering a unified framework for time series modeling. By incorporating higher-order graph aggregation through $\kappa$, Fighter achieves a multi-scale understanding of temporal dependencies, naturally expressed as graph relationships. The heatmaps corroborate Fighter's ability to capture both local and global dependencies, providing a mechanistically interpretable view of its predictions. These insights, validated by competitive performance on standard forecasting benchmarks, highlight Fighter's potential as a powerful and interpretable architecture for time series modeling and beyond.

## C  ADDITIONAL EXPERIMENT

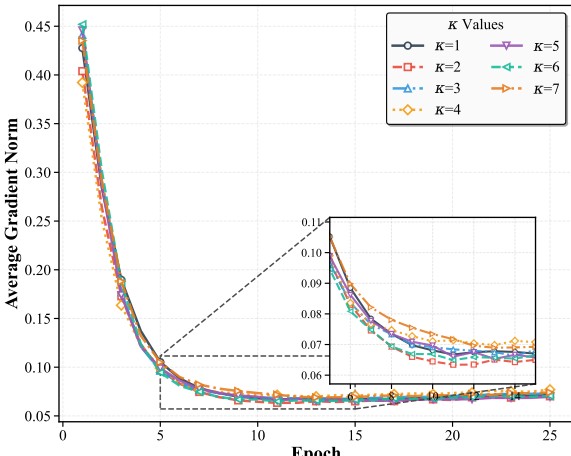

Figure 6: The training average gradient norm curves across different $\kappa$.

In Table 6, we record the GPU memory usage comparison for Fighter ($\kappa = 3$) and the baselines on the Weather dataset when the prediction length is 96. Fighter ($\kappa = 3$) implements the multi-hop graph aggregation, which enables long-range information to propagate naturally by higher-order compositions of the attention distribution matrix. As expected, it consumes relatively more GPU memory than the baselines but obtains superior performance. Furthermore, when $\kappa = 1$, Fighter exactly recovers a simplified version of the vanilla Transformer in which (i) the attention distribution matrix effectively collapses to the identity matrix, and (ii) the redundant second linear projection of the feed-forward network is removed (as justified by our GCN equivalence). Consequently, Fighter ($\kappa = 1$) consumes noticeably less GPU memory than the vanilla Transformer. This confirms that the extra memory cost of Fighter is incurred only by the multi-hop aggregation branches ($\kappa \geq 1$) and that, even at $\kappa = 1$, our streamlined design already brings a small but non-trivial memory saving without any performance degradation. Notably, the overhead is linear in $\kappa$, as presented

in Table 7. Although Fighter($\kappa = 3$) introduces additional computational branches, resulting in a moderate increase in GPU memory usage, the overall consumption remains lightweight and well within practical limits. This overhead is justified by the substantial performance gains Fighter delivers compared with conventional Transformer-based models.

During training, we also record the average gradient norm comparsion of Fighter across different $\kappa$ in Figure 6. As demonstrated, the average gradient norm of Fighter across varying $\kappa$ exhibits a vivid trend toward optimization stability: it starts at a relatively high value in the early training epochs and gradually decreases to a small and stable range as training proceeds. Importantly, the introduction of $\kappa$ does not disrupt the stability of the optimization dynamics. The gradient norms of Fighter($\kappa = 3$) follows a similar decay pattern and maintain comparable fluctuation levels to those corresponding to other $\kappa$ values. This indicates that the $\kappa$ does not introduce gradient explosion or vanishing phenomena, meanwhile, does not lead to abnormal oscillations or instability during training.

Table 6: GPU memory usage for Fighter ($\kappa = 3$) and different baselines on the Weather dataset (the prediction length is 96).

| Model | Autoformer | Informer | Reformer | Transformer | Fighter($\kappa = 3$) |
|---|---|---|---|---|---|
| **GPU Mem (MB)** | 108.0 | 120.2 | 105.6 | 109.1 | 157.3 |

Table 7: GPU memory usage for Fighter ($\kappa \in [1, 7]$) on the Weather dataset (the prediction length is 96).

| Model | $\kappa = 1$ | $\kappa = 2$ | $\kappa = 3$ | $\kappa = 4$ | $\kappa = 5$ | $\kappa = 6$ | $\kappa = 7$ |
|---|---|---|---|---|---|---|---|
| **GPU Mem (MB)** | 94.5 | 125.4 | 157.3 | 188.8 | 233.2 | 278.7 | 337.1 |

Table 8: Visualization of attention heatmaps for the first 64 keys, comparing **Fighter** and Transformer. Rows indicate the models; columns represent training epochs (25, 50, 75, 100).

| **25** | **50** | **75** | **100** |
|:---:|:---:|:---:|:---:|

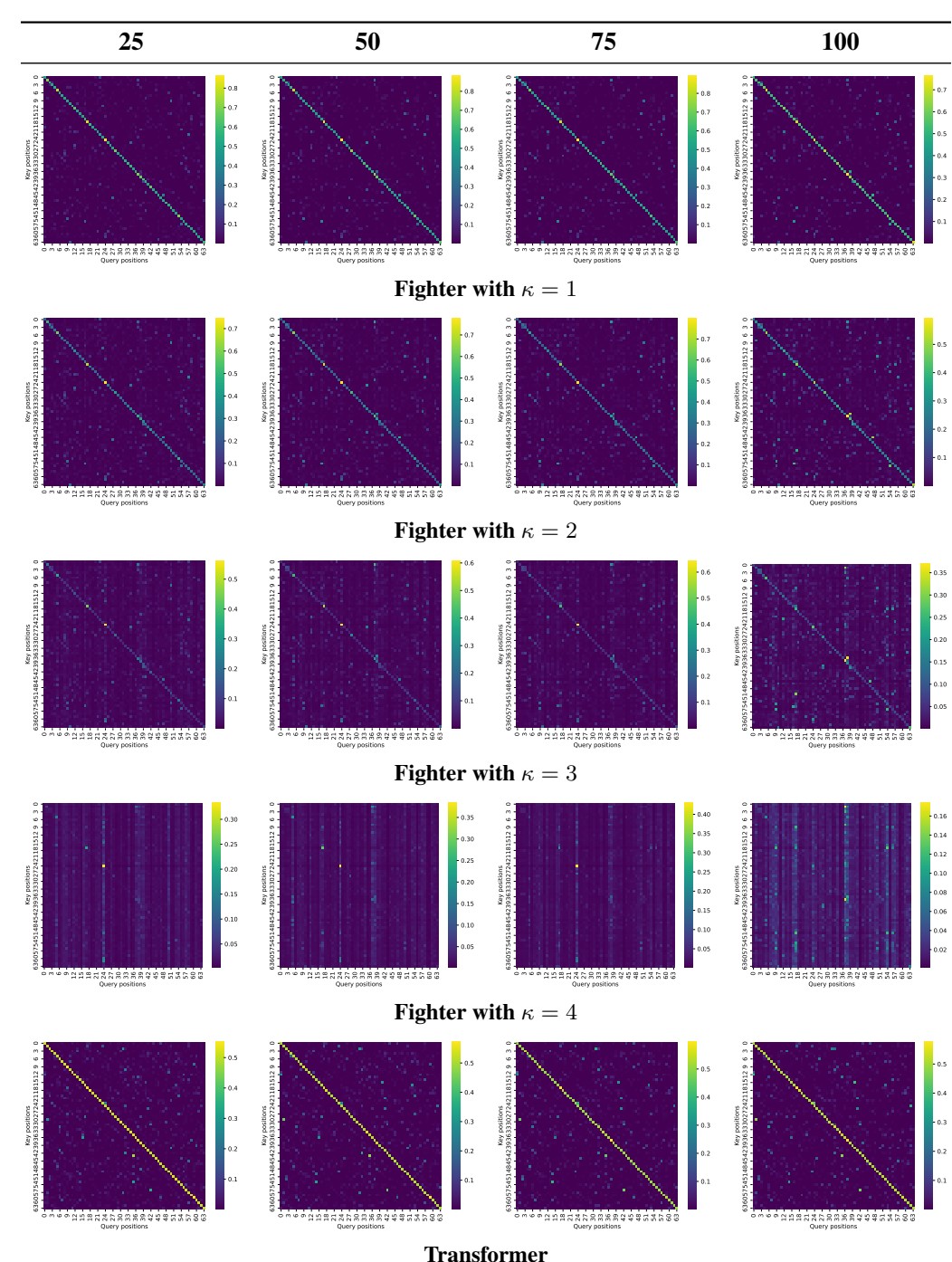

**Fighter with** $\kappa = 1$

**Fighter with** $\kappa = 2$

**Fighter with** $\kappa = 3$

**Fighter with** $\kappa = 4$

**Transformer**