# OpenReview forum: "Fighter: Unveiling the Graph Convolutional Nature of Transformers in Time Series Modeling"
_ICLR.cc/2026/Conference — Submitted to ICLR 2026_

### Official Review · Reviewer_RRhp · 2025-10-20

**Soundness:** 1
**Presentation:** 1
**Contribution:** 1
**Rating:** 2
**Confidence:** 4

**Summary:**

The paper explores analogies between Transformers and GNNs in the context of time series forecasting, drawing parallels between the attention mechanism and the message-passing framework. However, the paper has very limited technical novelty, some claims are overstated, and the empirical analysis has significant issues.

**Strengths:**

* Understanding attention-based architectures in the context of time series forecasting is an important and timely research problem.

**Weaknesses:**

* **Limited technical novelty.**  The technical novelty of the paper is very limited.
    - The paper’s main premise , i.e., that Transformers can be seen as a specific implementation of a GNN operating on a fully connected graph with data-dependent edge weights, is an obvious and well-known fact, which has been known and discussed in the community for years (see, e.g., [1] and the popular 2020 blog post [2]).
    - The MLP layer used between attention blocks in Transformer architectures increases model capacity and transforms the representations extracted by the attention block. The paper argues that this transformation is redundant since it is not used in vanilla GNN architectures (see, e.g., lines 199–201). However, there is no logical connection between the fact that a layer is not used in vanilla GNNs and it being generally unnecessary.
    - Given that the information propagation mechanism is the same, the equivalence observed when analyzing derivatives (Section 4.2) appears obvious and does not provide any novel insight.
    - The paper claims that, inspired by the analogy between GNNs and Transformers, one can use powers of the learned adjacency matrix to propagate information. However, since the graph is fully connected and the weighting mechanism data-adaptive, it is unclear what advantage this offers over standard multi-head attention.

* **Poor empirical evaluation.**  There are several issues in the empirical evaluation
    - The proposed approach is evaluated only against outdated baselines, on a small number of datasets, and with no indication of variability (e.g., standard deviations across runs). Moreover, the paper does not clearly describe how baselines were trained and tuned. This, combined with the absence of released code, makes the results difficult to reproduce.
    - The reported performance gains appear unreasonable (e.g., nearly 100× on some datasets) given the minor architectural modifications introduced. Furthermore, based on the sensitivity analysis, one would expect Fighter with a single hop to perform similarly to a Transformer, as the architectures are essentially equivalent. However, the reported performance on Weather is ~100× better even for a single hop, compared to the Transformer results in the table - while the Transformer performance shown in the plot (red line) appears more in line with Fighter. What is happening here? Are you sure the results were reported on the correct scale?  In addition, forecast accuracy on the Electricity dataset seems to improve with longer forecasting horizons, which is quite implausible since the task becomes more difficult.

Even disregarding the empirical issues, the very limited technical novelty alone would prevent me from recommending acceptance.


References

[1] Dwivedi et al., "A Generalization of Transformer Networks to Graphs", arxiv 2020

[2] Chaitanya Joshi, "Transformers are Graph Neural Networks*", 2020 blog post: https://graphdeeplearning.github.io/post/transformers-are-gnns/ -- (arxiv 2025)

**Questions:**

See weaknesses.

---

> ### Author Response · Authors · 2025-11-29
>
> # Part (1/2)
>
> Thank you for your detailed review and valuable feedback. We are encouraged that you found our exploration of the Transformer-GNN connection in time series forecasting to be "an important and timely research problem." Below, we provide a point-by-point response to the concerns raised. We hope our responses clarify any concerns and provide the necessary explanation.
>
> > **[W1.1]** On Technical Novelty & Relation to Prior Work
>
> While the high-level view of attention as a graph is known, our work provides the first systematic, mechanistic unification of Transformer encoders and GCNs, leading to novel architectural insights.
>
> - Beyond a High-Level Analogy: Unlike prior works [1, 2] which stop at the forward-pass analogy, we establish a comprehensive equivalence by rigorously analyzing both the forward pass (Sec 4.1) and the backward pass (Sec 4.2). We are the first to derive and compare the gradient updates for the feature transformation parameters (W_V, W_FFN) and, crucially, for the dynamic structure-learning parameters (W_Q, W_K), demonstrating how the "graph" is dynamically learned from data, unexplored in this context.
>
> - From Theory to Architecture: This theoretical analysis is not just exposition; it directly exposes a redundancy. We are the first to identify that the post-attention MLP projection W_FFN^{(\ell,2)} (justified in Sec 4.1) is functionally subsumed by subsequent layers from a GCN perspective. This insight led to the novel, streamlined design of Fighter, which is a concrete architectural contribution beyond prior conceptual discussions [1,2].
>
> > **[W1.2]** On the Necessity of the MLP Layer
>
> Thank you for your comment. It seems there might have been some misunderstanding. Our argument is not that the MLP (FFN) is universally redundant, but that the specific, second linear projection (W_FFN^(ℓ,2)) in the standard Transformer block is redundant from the perspective of our GCN equivalence. Fighter does not remove feature transformation entirely. It streamlines the block by consolidating the value projection and the first FFN projection into a single transformation W^(ℓ). This projection (W_FFN^(ℓ,2)), while harmless for performance in NLP, obscures interpretability in time series: it linearly mixes the post-attention features before the next layer, breaking the clean separation between “temporal modeling” (attention) and “spatial transformation,” making it impossible to directly interpret the attention matrix as the sole driver of temporal dependencies.

---

> > ### Author Response · Authors · 2025-11-29
> >
> > # Part (2/2)
> >
> > > **[W1.3]** On the Obviousness of the Derivative Analysis
> >
> > Thank you for the comment. We would like to emphasize that the core novelty of the backward-pass analysis lies in Equations 8 and 9, which have not appeared in any prior Transformer–GNN analogy work (including [1], [2], or subsequent papers).
> >
> > These two gradient formulas are the first explicit derivation of how the query and key matrices (W_Q, W_K) are updated to learn a fully data-dependent, dynamic adjacency matrix. Standard GCNs have no analogous mechanism because their adjacency matrix is fixed or pre-defined; there is simply no gradient flow that adapts the graph structure itself. By deriving the exact form of \nabla W_Q and \nabla W_K, we reveal the precise learning dynamics that make attention spatially adaptive at every training step, a property that is crucial for time series modeling (where the strength and range of temporal dependencies vary strongly across channels and instances).
> >
> > This structure-learning perspective explains mechanistically why Transformers can discover irregular, instance-specific temporal patterns that static-graph or hand-crafted convolution methods cannot. In the time series context, it offers a new lens for understanding how spatial–temporal interactions are jointly optimized, going far beyond the forward-pass similarity discussed in earlier works.
> >
> > [1] Dwivedi VP, Bresson X. "A generalization of transformer networks to graphs." arXiv preprint arXiv:2012.09699 (2020).
> >
> > [2] Joshi CK. “Transformers are graph neural networks.” arXiv preprint arXiv:2506.22084. 2025 Jun 27.
> >
> > > **[W1.4]** On Multi-Hop vs. Multi-Head Attention
> >
> > Multi-hop aggregation and multi-head attention are fully orthogonal and serve cleanly separated roles in the temporal vs. spatial dimensions:
> >
> > - Multi-Hop operates exclusively on the temporal axis: it strengthens and extends pure temporal propagation by concatenating features from 1-hop, 2-hop, …, κ-hop neighbors in the dynamic graph defined by the attention matrix ([I, smx, smx^2, \dots, smx^{\kappa-1}]). This yields explicit, controllable, and interpretable long-range temporal paths without any additional channel mixing..
> >
> > - Multi-Head attention operates exclusively on the spatial/channel axis: at each hop distance, different heads learn independent linear projections of the same temporally aggregated features, allowing the model to attend to different channel subspaces (analogous to ensemble diversity in MixHop [3], but now within a single, coherent temporal graph).
> >
> > Because Fighter keeps temporal propagation (powers of the attention matrix) strictly separated from spatial transformation (the single shared W^ℓ applied after temporal aggregation), combining both mechanisms is natural and highly effective: multi-hop gives precise control and interpretability over temporal receptive field and multi-scale dependencies, while multi-head further boosts spatial expressiveness at every hop level. In our implementation, multi-head attention is fully integrated on top of multi-hop aggregation with no conflict.
> >
> > This clean temporal–spatial separation is a direct consequence of removing the second FFN projection (W1.2) and is not possible in the standard Transformer block, where post-attention channel mixing immediately blurs the two dimensions.
> >
> > [3] Abu-El-Haija S, Perozzi B, Kapoor A, Alipourfard N, Lerman K, Harutyunyan H, Ver Steeg G, Galstyan A. "Mixhop: Higher-order graph convolutional architectures via sparsified neighborhood mixing." ICML 2019.
> >
> > > **[W2]** On Baselines, Variability, and Reproducibility
> >
> > Please refer to the **Addressing Limitations in Dataset Coverage, Forecasting Horizons, and Statistical Reporting** & **Generality of Multi-Hop Graph Aggregation: Improvements on Recent Baselines** section in the common questions comment.

---

### Official Review · Reviewer_SazB · 2025-10-29

**Soundness:** 3
**Presentation:** 4
**Contribution:** 3
**Rating:** 6
**Confidence:** 3

**Summary:**

The work focuses on proving the equivalence between a Transformer encoder and a flexible graph convolution network.
The authors also propose a flexible graph convolution transformer called Fighter, which removes redundant linear projections to improve efficiency and claims to have higher expressivity.

**Strengths:**

The paper is written clearly, and theoretical results are presented briefly in the main paper with further details deferred to the appendix. I also like the illustrations to aid the readers.

**Weaknesses:**

I find the paper interesting and well-written overall. However, the experimental results section could be strengthened. I have included several questions and suggestions in that regard. I'm happy to improve the rating if my questions are answered.

**Questions:**

1. What is the difference in the attention matrices of the Transformer and the GCN? Could you please report the norm of the difference? Basically, Table 5 shown in numbers.
2. In Table 1, why is there a huge drop in MSE/MAE in the forecasting tasks, but comparatively a minimal improvement in the classification accuracy?
3. Can the authors please provide a table comparing the number of parameters of different baseline models with Fighter, and also their per-epoch training time?
4. Can the authors please provide confidence intervals for all the numerical results?
5. Commenting on the better performance of Fighter over longer sequences just by comparing two datasets is not wise. To make such a claim, the trend must be observed over a larger number of data points (in this case, datasets and prediction length).
6. Fig 4(b) is not clear. Could the authors please zoom in or adjust the scale so that the order over different $\kappa$ values can be observed?

---

> ### Author Response · Authors · 2025-11-29
>
> Thank you for the detailed and constructive feedback. We appreciate your recognition of the paper’s clarity, concise theoretical presentation, helpful illustrations, and the novel equivalence between Transformers and GCNs, as well as the efficiency and interpretability gains from Fighter. We are glad to address your questions on the experimental section to strengthen it further. Below, we respond to each point in order.
>
> > **[Q1]** Regarding the difference in attention matrices between the Transformer and GCN (presumably referring to Fighter)
>
> Both use the same base attention distribution via softmax(d^{-1/2}QK^\top), but Fighter extends this to multi-hop aggregation with smx^[κ] = [I, smx, smx², ..., smx^(κ-1)] to capture longer-range dependencies directly rather than relying on the implicit stacking of single-hop layers in the vanilla Transformer. This extension significantly enriches the receptive field within each layer. To quantify the effect, we computed the Frobenius norm of the difference between the Transformer's single-hop attention matrix and the multi-hop matrix of Fighter ($\kappa$=3): 5.31. The norm reflects the added expressivity from higher-order terms.
>
> > **[Q2]** On the drop in MSE/MAE for forecasting versus minimal improvement in classification accuracy in Table 1
>
> The metrics are inherently incomparable—large MSE/MAE reductions in forecasting do not translate to equivalent accuracy gains in classification due to different scales and sensitivities. Forecasting errors (MSE/MAE) are continuous and directly reflect dependency modeling, where Fighter’s multi-hop aggregation yields substantial improvements on Electricity and Weather. In contrast, AG News [1] classification operates on discrete, high-level semantic patterns, where even strong dependency capture leads to modest accuracy gains. This behavior aligns with dataset characteristics, with time series tasks benefiting more from explicit long-range modeling. We will clarify this in the revision.
>
> [1] Zhang X, Zhao J, LeCun Y. "Character-level convolutional networks for text classification." NeurIPS 2015.
>
> > **[Q3]** Comparing the number of parameters and per-epoch training time of different baseline models with Fighter,
>
>
> Please refer to the **Clarification and Empirical Analysis of Computational and Memory Complexity** section in the common questions comment.
>
> > **[Q4-5]** Confidence intervals for Table 1 results & Evaluation on large dataset
>
> Please refer to the **Addressing Limitations in Dataset Coverage, Forecasting Horizons, and Statistical Reporting** section in the common questions comment.
>
> > **[Q6]** Zoom in Fig 4(b)
>
> We apologize for the lack of clarity. We have updated the figure in the revision.

---

### Official Review · Reviewer_rpL3 · 2025-10-31

**Soundness:** 2
**Presentation:** 2
**Contribution:** 2
**Rating:** 4
**Confidence:** 4

**Summary:**

The authors focus on time series forecasting and introduce a connection between the transformer encoder and the graph convolutional network, by mapping the attention distribution matrix to a dynamic adjacency matrix, the layer transformations to graph convolution, and the updated attention values and projection coefficients to the GCN parameters. Finally, based on this analogy, the Flexible Graph Convolutional Transformer, so-called Fighter, is proposed, excluding unnecessary linear projections and considering multi-hop graph aggregation, representing multi-scale temporal dependencies by graph edges.

**Strengths:**

1. **Originality:** The authors make an analogy between the graph convolution and the transformer encoders in time series forecasting, and mathematically prove the connection under specific circumstances, which is a novel approach bringing two hot modeling aspects in modern time series frameworks.
2. **Quality and Clarity:** The methodological section (and additional details in the appendix) is easy to follow, despite being heavy in formulas.
3. **Significance:** On the few showcased datasets, the forecasting performance of the proposed Fighter model significantly outperforms transformer-based baselines.

**Weaknesses:**

- **W1-Significance (Motivation of Proposed Contribution):** The motivation behind the proposed idea is not clearly showcased. The authors aim to connect graph convolution representations with transformer attention for time series forecasting, yet it is not clearly explained why existing approaches are inadequate. In particular, several transformer variants addressing optimization issues in time series forecasting have already been proposed, but the paper does not clarify whether these methods are conceptually related to the proposed approach or whether the limitations they face are effectively addressed by the new method.
- **W2-Clarity (Positioning against Related Works):** The authors do not discuss related work in graph representation learning for time series, where methods often distinguish between sparse [1,3] vs. fully-connected adjacencies [4] and static vs. dynamic dependencies [4,5]. The Introduction and Related Work sections should be updated to position the proposed Fighter model in the context of existing graph-based time series approaches. Importantly, the work seems conceptually similar to StemGNN [5], where the adjacency is learned based on a latent correlation attention layer on the whole sequence.
- **W3-Quality and Significance (Experimental Evaluation):** The set of baselines considered is limited to older transformer-based approaches and does not include more recent improvements, alternative architectures, or graph-based methods (see references and TimeMixer, TimeXer, iTransformer, and PatchTST from https://github.com/thuml/Time-Series-Library). Additionally, the datasets used do not cover the full benchmark in the forecasting community (e.g., several standard datasets are not evaluated, and few horizon lengths are considered). The inclusion of the text classification dataset is unclear, as it is not standard in the time series community and does not directly relate to the forecasting task. If additional tasks are to be considered, they should be standard in the time series field, such as classification and anomaly detection, to ensure comparability with existing literature.
- **W4-Clarity and Significance (Computational Complexity):** Although the authors refer to computational improvements enabled by their proposed method, they do not provide a computational analysis or experimental comparison for time cost and memory complexity compared to baselines or variants of the model. The fully-connected design of the adjacency, combined with message passing, is in general computationally expensive and should be compared to baselines to justify the claims for computational improvement.
W5-Significance (Reproducibility): Although a substantial explanation of the experimental setup is given, the code implementation of the proposed method is not available in the submission; therefore, direct reproducibility of the experimental results is not possible.

**References:**
1. Shang C, Chen J, Bi J. Discrete graph structure learning for forecasting multiple time series. arXiv preprint arXiv:2101.06861. 2021 Jan 18.
2. Bai L, Yao L, Li C, Wang X, Wang C. Adaptive graph convolutional recurrent network for traffic forecasting. Advances in neural information processing systems. 2020;33:17804-15.
3. Wu Z, Pan S, Long G, Jiang J, Chang X, Zhang C. Connecting the dots: Multivariate time series forecasting with graph neural networks. InProceedings of the 26th ACM SIGKDD international conference on knowledge discovery & data mining 2020 Aug 23 (pp. 753-763).
4. Yi K, Zhang Q, Fan W, He H, Hu L, Wang P, An N, Cao L, Niu Z. FourierGNN: Rethinking multivariate time series forecasting from a pure graph perspective. Advances in neural information processing systems. 2023 Dec 15;36:69638-60.
5. Cao D, Wang Y, Duan J, Zhang C, Zhu X, Huang C, Tong Y, Xu B, Bai J, Tong J, Zhang Q. Spectral temporal graph neural network for multivariate time-series forecasting. Advances in neural information processing systems. 2020;33:17766-78.

**Questions:**

1. Can the authors showcase with examples/illustrations how the proposed design solves long-term problems in the time series community, e.g., correlations, dynamic dependencies, distribution shift, or other?
2. Can the authors position clearly the proposed method against related works in temporal modeling and graph-based modeling, including justifications for the architectural choices with respect to common challenges (see Q1)?
3. The experimental comparisons should be extended to more relevant baselines and datasets or more time series tasks to improve the significance of the contribution.
4. What are the computational aspects of the proposed architecture and how do these compare to baselines?

---

> ### Author Response · Authors · 2025-11-29
>
> # Part (1/4)
>
> Thank you for your detailed feedback and for recognizing the originality of our analogy between graph convolutions and Transformer encoders, the clarity of our methodological explanations, and Fighter's forecasting improvements over baselines on the showcased datasets. We appreciate your constructive suggestions and address each point below. We hope that our response resolves your concerns.
>
> > **[W1]** Significance (Motivation)
>
> Although recent Transformer variants have improved efficiency, they still inherit the vanilla Transformer’s opaque internal mechanics [1,2] and retain a redundant linear projection after each FFN (the second projection in the two-layer MLP). This projection, while harmless for performance in NLP, obscures interpretability in time series: it linearly mixes the post-attention features before the next layer, breaking the clean separation between “temporal modeling” (attention) and “spatial transformation,” making it impossible to directly interpret the attention matrix as the sole driver of temporal dependencies. As a result, even advanced models remain black boxes despite strong performance (e.g., Autoformer's [3] decomposition improves accuracy by up to 38% on benchmarks, yet retains opaque internals and redundant projections that inflate overhead without enhancing interpretability.)
>
> Graph-based methods, while effective for structured dependencies (e.g., in spatio-temporal tasks like traffic forecasting via models such as STGCN [4]), rely on predefined static adjacencies that fail to adapt to dynamic temporal shifts in univariate or multivariate series. This inadequacy is evident in their limited applicability to irregular or evolving patterns, as noted in integrations like GTA [5] for anomaly detection, where graph convolutions are combined with Transformers but without a unified theoretical foundation.
>
> Our Fighter addresses these gaps by establishing a mathematical equivalence between Transformer encoders and GCNs, where the attention matrix acts as a dynamic adjacency for feature aggregation. This not only eliminates redundant linear projections (subsumed by subsequent layers in the GCN view) but also enables multi-hop aggregation to explicitly model multi-scale dependencies, as interpretable graph edges. Unlike prior Transformer variants that patch efficiency issues without demystifying internals, Fighter provides mechanistic insights: attention matrices can be visualized as graphs, revealing how long-range correlations are captured. Empirically, this leads to superior forecasting, with MSE/MAE reductions of 14.0%/6.8% on Electricity-720 compared to vanilla Transformer.
>
> [1] Lim B, Arık SÖ, Loeff N, Pfister T. "Temporal fusion transformers for interpretable multi-horizon time series forecasting." International journal of forecasting 37.4 (2021): 1748-1764.
>
> [2] Wen Q, Zhou T, Zhang C, Chen W, Ma Z, Yan J, Sun L. "Transformers in time series: a survey." IJCAI. 2023.
>
> [3] Wu H, Xu J, Wang J, Long M. "Autoformer: Decomposition transformers with auto-correlation for long-term series forecasting." NeurIPS. 2021.
>
> [4] Yu B, Yin H, Zhu Z. "Spatio-Temporal Graph Convolutional Networks: A Deep Learning Framework for Traffic Forecasting." IJCAI, 2018.
>
> [5] Chen Z, Chen D, Zhang X, Yuan Z, Cheng X. “Learning graph structures with transformer for multivariate time-series anomaly detection in IoT.” IEEE Internet of Things Journal. 2021 Jul 27;9(12):9179-89.

---

> > ### Author Response · Authors · 2025-11-29
> >
> > # Part (2/4)
> >
> > > **[W2]** Clarity (Positioning against Related Works)
> >
> > We thank the reviewer for this valuable suggestion and will significantly expand the Introduction and Related Works sections in the revision to clearly position Fighter within the graph-based time series literature, using the widely accepted spatial-vs-temporal modeling lens.
> > Most existing graph-based forecasting models treat multivariate channels as nodes (spatial dimension) and construct graphs primarily over this spatial dimension:
> >
> > - GTS [6], MTGNN [7], AGCRN [8], and StemGNN [10] all learn an adjacency matrix across variables (inter-series correlations), while temporal dynamics are handled separately via dilated convolutions, recurrence, or spectral temporal kernels. Their graphs are either static [6,8], node-adaptive but still fixed per sample [7], or defined in the spectral domain [10]. None of them model the temporal dimension itself as a graph.
> >
> > - FourierGNN [9] is a notable exception: it flattens both variables and time steps into a single hyper-node set, building a fully-connected space-time graph and performing convolution jointly over spatial and temporal dimensions. While powerful in theory, this incurs prohibitive O((N×T)²) cost, forcing the use of sliding windows that severely limit receptive field and long-range temporal modeling.
> >
> > Fighter takes a fundamentally different perspective: it treats the time steps themselves as nodes and uses the attention distribution matrix as a fully-connected, input-adaptive graph over the temporal dimension. Inter-variable (spatial) dependencies are naturally captured by the feature extraction layers (streamlined value and feed-forward projection), which operate on the channel dimension similarly as in Transformers. This clean separation yields two critical advantages:
> >
> > - Temporal modeling becomes fully dynamic and attention-driven (no static or predefined temporal edges), while remaining quadratic only in sequence length T (not N×T), preserving the efficiency against FourierGNN [9].
> >
> > - Multi-hop aggregation (\kappa\geq3) explicitly strengthens temporal propagation without mixing spatial and temporal graphs, enabling controllable, interpretable long-range temporal paths. Multi-head attention, when used, further enhances spatial expressiveness by allowing different heads to focus on distinct channel subspaces at each hop level—analogous to multiple random initializations in MixHop but applied within the same efficient temporal graph framework.
> >
> > Thus, Fighter is the first model to cast pure temporal dependencies as a dynamic, learnable graph derived from attention, sidestepping the computational explosion of joint space-time graphs [9] and the rigidity of spatial-only graphs [6,7,8,10], while inheriting Transformer efficiency and adding explicit multi-scale temporal interpretability through graph edges.
> >
> > [6] Shang C, Chen J, Bi J. "Discrete Graph Structure Learning for Forecasting Multiple Time Series." ICLR. 2021.
> >
> > [7] Wu Z, Pan S, Long G, Jiang J, Chang X, Zhang C. "Connecting the dots: Multivariate time series forecasting with graph neural networks." KDD. 2020.
> >
> > [8] Bai L, Yao L, Li C, Wang X, Wang C. "Adaptive graph convolutional recurrent network for traffic forecasting." NeurIPS 2020.
> >
> > [9] Yi K, Zhang Q, Fan W, He H, Hu L, Wang P, An N, Cao L, Niu Z. "FourierGNN: Rethinking multivariate time series forecasting from a pure graph perspective." NeurIPS 2023.
> >
> > [10] Cao D, Wang Y, Duan J, Zhang C, Zhu X, Huang C, Tong Y, Xu B, Bai J, Tong J, Zhang Q. "Spectral temporal graph neural network for multivariate time-series forecasting." NeurIPS 2020.
> >
> > > **[W3]** Quality and Significance (Experimental Evaluation):
> >
> > Please refer to the **Clarification and Empirical Analysis of Computational and Memory Complexity** & **Generality of Multi-Hop Graph Aggregation: Improvements on Recent Baselines** section in the common questions comment.
> >
> >
> > > **[W4]** Clarity and Significance (Computational Complexity):
> >
> > Please refer to the **Clarification and Empirical Analysis of Computational and Memory Complexity** section in the common questions comment.
> >
> >
> > > **[W5]** Significance (Reproducibility):
> >
> > We have prepared code for release, including full implementations, hyperparameters, and scripts for all experiments.

---

> > > ### Author Response · Authors · 2025-11-29
> > >
> > > # Part (3/4)
> > >
> > > > **[Q1]** How the proposed design solves long-term problems in the time series community
> > >
> > > Fighter directly tackles core time series challenges by reinterpreting the Transformer encoder as a graph convolutional network over the temporal dimension, where time steps are treated as nodes, the attention distribution matrix serves as a fully dynamic and input-adaptive adjacency matrix (learned end-to-end via query-key interactions), and the simplified value + FFN projections handle feature extraction across channels (spatial/multivariate dimension). This clean separation avoids the prohibitive O((N×T)²) cost of jointly graphing spatial and temporal dimensions (as in FourierGNN [9]), while preserving Transformer-level efficiency.
> > >
> > > - Long-range correlations & periodicity: Multi-hop aggregation (κ\geq 3) explicitly propagates information along the temporal graph by raising the attention-derived adjacency to the κ-th power, creating direct paths between distant timestamps.
> > >
> > > - Dynamic dependencies: Because the adjacency is derived from attention, edge weights adapt per input sample. During anomalous weather events (e.g., sudden cold fronts), query-key similarity automatically strengthens links between affected variables and timestamps, enabling context-aware modeling that static or predefined graphs cannot achieve.
> > >
> > > Thus, Fighter’s graph-convolutional view over the temporal dimension, with attention-driven dynamic edges, streamlined spatial feature extraction, and controllable multi-hop propagation, directly translates the theoretical equivalence into concrete solutions for long-range, dynamic, and non-stationary dependencies in real-world time series.
> > >
> > > > **[Q2]** Position Fighter against related works
> > >
> > > Fighter positions itself distinctly against both temporal (Transformer-based) and graph-based models by leveraging the proven equivalence between attention and dynamic graph convolution.
> > >
> > > Vs. temporal models: Standard Transformers and their efficient variants model dependencies implicitly via single-hop attention and handle channels through heavy value + two-layer FFN projections. While powerful, they (i) lack explicit control over temporal propagation depth, making long-range and hierarchical periodicities hard to interpret and stabilize (challenge: long-range correlations & periodicity), and (ii) retain a redundant output projection in the FFN that breaks the clean separation between temporal learning and feature transformation, reducing mechanistic clarity.
> > >
> > > Fighter removes this redundant projection (justified and proven subsumed by the next layer in the GCN view, Eq. 10) and replaces implicit single-hop propagation with explicit multi-hop aggregation on a temporal graph (κ≥3). This directly strengthens long-range and multi-scale temporal modeling while preserving exact Transformer efficiency.
> > >
> > > Vs. graph models: Most graph forecasting models [6,7,8,10] construct graphs only over the spatial (multivariate) dimension and treat time with separate mechanisms. This works well for fixed inter-variable relations but cannot adapt temporal connectivity per sample and struggles with purely temporal challenges (dynamic dependencies & distribution shifts).
> > > FourierGNN [9] is the only prior work that graphs both dimensions jointly, but the resulting O((N×T)²) cost forces restrictive sliding windows, effectively crippling true long-range temporal modeling.
> > >
> > > Fighter flips the perspective: it graphs only the temporal dimension (time steps as nodes), derives a fully dynamic, input-adaptive adjacency directly from attention (no predefined or sampled edges), and delegates clean, efficient channel-wise modeling to the streamlined value + FFN path. Multi-head attention further allows different heads to capture distinct channel subspaces at each hop level (analogous to multiple random initializations in MixHop, but on a pure temporal graph). This design (i) avoids the computational explosion of joint space-time graphs, (ii) gives fully adaptive temporal edges for dynamic dependencies and shifts, (iii) provides explicit multi-hop control for hierarchical periodicity and long-range stability, and (iv) yields interpretable temporal relations as visualizable graph edges.
> > >
> > > In summary, Fighter is the first architecture to treat the temporal axis itself as a dynamic, attention-learned graph, achieving the interpretability and multi-scale expressiveness that spatial-only graph models and implicit-attention Transformers lack, without paying the prohibitive price of joint space-time graph convolution. These choices are theoretically grounded in the Transformer - GCN equivalence and empirically validated by consistent superiority on standard benchmarks (see extended results in the revision).

---

> > > > ### Author Response · Authors · 2025-11-29
> > > >
> > > > # Part (4/4)
> > > >
> > > > > **[Q3]** Experimental comparisons
> > > > Please refer to the **Clarification and Empirical Analysis of Computational and Memory Complexity** & **Generality of Multi-Hop Graph Aggregation: Improvements on Recent Baselines** section in the common questions comment.
> > > >
> > > >
> > > > > **[Q4]** What are the computational aspects of the proposed architecture and how do these compare to baselines?
> > > >
> > > > Please refer to the **Clarification and Empirical Analysis of Computational and Memory Complexity** section in the common questions comment.

---

### Official Review · Reviewer_HwzJ · 2025-11-01

**Soundness:** 3
**Presentation:** 2
**Contribution:** 2
**Rating:** 4
**Confidence:** 4

**Summary:**

This paper presents FIGHTER (Flexible Graph Convolutional Transformer), a novel framework that reinterprets Transformers as Graph Convolutional Networks (GCNs) for time series modeling. The authors provide a unified theoretical analysis showing that the Transformer’s attention distribution acts as a dynamic adjacency matrix, while its value and feed-forward updates mimic GCN feature propagation. Building on this equivalence, they propose FIGHTER, a simplified Transformer variant that removes redundant projections and introduces multi-hop graph aggregation, enabling explicit, interpretable temporal dependencies. Experiments on standard forecasting datasets (Electricity, Weather) and text classification (AG News) show that FIGHTER achieves state-of-the-art or superior results with dramatically lower errors and improved interpretability through graph-based attention visualizations.

**Strengths:**

The paper makes an important conceptual contribution by rigorously connecting the Transformer and GCN formulations, offering both theoretical insight and architectural innovation. The idea of treating attention as a learnable adjacency matrix provides a fresh and unifying lens for understanding sequence modeling. The derivations are detailed and mathematically grounded, bridging the forward and backward pass analysis of both architectures. Empirically, FIGHTER achieves substantial performance gains and offers clear interpretability via graph-based attention visualization. The design is also elegant—removing redundant projections and adding multi-hop aggregation leads to improved efficiency and better long-range dependency capture.

**Weaknesses:**

The experiments, though diverse, are somewhat limited in dataset variety and task complexity;

Additional real-world datasets or ablations on larger Transformer variants would strengthen the empirical foundation.

Moreover, the mathematical exposition is dense and sometimes overly formal, which might obscure intuition for non-theoretical readers.

The practical computational trade-offs of the multi-hop attention (in memory and speed) are not deeply discussed.

Finally, interpretability is primarily visual and qualitative; a more quantitative assessment of interpretive fidelity would improve credibility.

**Questions:**

How does FIGHTER scale computationally with increasing hop parameter κ, especially in long sequences?

Can the proposed equivalence framework generalize to multi-head or cross-attention Transformers (e.g., encoder-decoder setups)?

Are there scenarios where the dynamic adjacency interpretation breaks down—for instance, in sparse attention or masked modeling settings?

How stable are the gradients and convergence behavior compared to standard Transformers during long training runs?

---

> ### Author Response · Authors · 2025-11-29
>
> # Part (1/2)
>
> Thank you to the reviewers for their thoughtful feedback and for recognizing the core contribution: the rigorous theoretical equivalence between Transformers and GCNs, the elegant design of Fighter via removing redundant projections and adding multi-hop aggregation, and the clear interpretability enabled by graph-based attention visualizations. Below, we respond to each point in order.
>
> > **[W1&W2]** Limited dataset variety and task complexity & ablations on larger Transformer variants
>
> Please refer to the **Addressing Limitations in Dataset Coverage, Forecasting Horizons, and Statistical Reporting** section in the common questions comment.
>
> We also conducted an additional ablation study on the number of Fighter layers (i.e., the depth of the Fighter). In all experiments reported below, each Fighter layer uses κ=3 as in Table 1.
>
> **Prediction horizon o=96**
> | Dataset | # Layers | MSE | MAE |
> | --- | --- | --- | --- |
> | Weather | 1 | 0.275 | 0.346 |
> | Weather | 3 | 0.258 | 0.330 |
> | Weather | 6 | **0.240** | **0.319** |
> | ETTh1 | 1 | 0.732 | 0.651 |
> | ETTh1 | 3 | 0.717 | 0.626 |
> | ETTh1 | 6 | **0.712** | **0.586** |
>
> The results clearly demonstrate that Fighter scales favorably with depth: increasing the number of layers from 1 to 6 consistently improves performance on both datasets. This confirms the good scaling property of our proposed Fighter.
>
> > **[W3]** Dense mathematical exposition
>
> We aimed for rigor in establishing the forward/backward equivalence, particularly the exact gradient derivations in Equations 8 and 9 for $W_Q$ and $W_K$. These formulas reveal, for the first time, precisely how query and key parameters learn a fully data-dependent dynamic adjacency matrix: something standard GCNs cannot do, because their adjacency is fixed and receives no gradients. This adaptive structural learning is critical for time series, where dependency patterns vary dramatically across channels and instances. To improve accessibility without compromising precision, we present a concise, intuitive overview at the beginning of Section 4.
>
> > **[W4]** Computational trade-offs of multi-hop
>
> Please refer to the **Clarification and Empirical Analysis of Computational and Memory Complexity** section in the common questions comment.
>
> > **[W5]** Interpretability primarily qualitative
>
> We conduct an ablation study on multi-hop aggregation to quantitively assess incorporating interpretability. Please refer to the **Generality of Multi-Hop Graph Aggregation: Improvements on Recent Baselines** section in the common questions comment. As shown in that Table, integrating this mechanism consistently improves the forecasting performance of iTransformer on the ETTh1 and ETTm1 datasets across short and long horizons, with clear reductions in both MSE and MAE. These gains confirm that the multi-hop aggregation serves as a general and effective enhancement for capturing long-range dependencies, even when applied to strong state-of-the-art baselines.
>
> > **[Q1]** Scaling with $\kappa$ on long sequences
>
> Please refer to the **Clarification and Empirical Analysis of Computational and Memory Complexity** section in the common questions comment.

---

> > ### Author Response · Authors · 2025-11-29
> >
> > # Part (2/2)
> >
> > > **[Q2]** Generalization to multi-head and cross-attention
> >
> > The theoretical equivalence holds cleanly in both multi-head and encoder-decoder settings, and multi-hop aggregation combines orthogonally with multi-head attention.
> >
> > - Multi-head: Each head computes its own attention distribution matrix, yielding head-specific dynamic adjacency matrices. Multi-hop aggregation is applied independently per head on the temporal axis ([I, smx_h, smx_h^2, …, smx_h^{κ-1}] for head h), while the subsequent linear projection remains shared across hops but can still be head-specific in the standard way. Thus, multi-hop controls explicit temporal receptive field and multi-scale propagation, whereas multi-head provides ensemble-like diversity in the channel/subspace dimension. The two mechanisms are completely decoupled and complementary: multi-hop extends temporal paths, multi-head enriches representation at every hop distance.
> >
> > - Cross-attention (encoder-decoder): The framework extends naturally. Self-attention in the encoder and decoder remains intra-sequence dynamic graphs as before. Cross-attention from decoder to encoder defines a bipartite dynamic adjacency from target tokens to source tokens, exactly analogous to a bipartite GCN layer. Multi-hop aggregation can be applied on this bipartite graph when longer-range source dependencies are needed. The encoder-only case we study is the foundational building block; all other attention types (cross, masked, sparse) inherit the same graph-convolutional interpretation.
> >
> > A brief discussion is presented in Remark 2.
> >
> > > **[Q3]** Breakdown in sparse or masked settings
> >
> > The interpretation is universal: any attention pattern (dense, sparse, masked) defines a valid dynamic adjacency. Sparse attention (e.g., Longformer [1]) corresponds to a sparse graph with fixed non-edges; masked positions zero out rows/columns, equivalent to node removal. The GCN equivalence holds locally within connected components.
> >
> > [1] Beltagy I, Peters ME, Cohan A. “Longformer: The long-document transformer.” arXiv preprint arXiv:2004.05150. 2020 Apr 10.
> >
> > > **[Q4]** Gradient stability and convergence
> >
> > In the revised manuscript, we have included **Figure 6**, which is the average gradient norm curves of Fighter under different values of κ during training. It shows that the average gradient norm of Fighter across varying $\kappa$ exhibits a clear trend toward optimization stability: it starts at a relatively high value in the early training epochs and gradually decreases to a small and stable range as training proceeds. Importantly, the introduction of $\kappa$ does not disrupt the stability of the optimization dynamics. The gradient norms of Fighter($\kappa=3$) follows a similar decay pattern and maintain comparable fluctuation levels to those corresponding to other $\kappa$ values. This indicates that the $\kappa$ does not introduce gradient explosion or vanishing phenomena, meanwhile, does not lead to abnormal oscillations or instability during training.

---

### Author Response · Authors · 2025-11-29
**Common Questions Part (1/2)**

We sincerely thank reviewers for their insightful and valuable feedback. First and foremost, we would like to apologize for the delayed submission of this rebuttal. While preparing our response, we discovered that the train/validation/test splits were not identical across different methods and ablations in the original experiments. We have immediately fixed the splitting procedure to ensure strictly identical splits for all methods, rerun all affected experiments, and are working intensively to provide the corrected results. All clarifications, additional analyses, ablation studies, and new results in this rebuttal will be included in the final version. Below, we first address common concerns raised by multiple reviewers, followed by responses to individual questions.

# Addressing Limitations in Dataset Coverage, Forecasting Horizons, and Statistical Reporting

We sincerely thank reviewers for pointing out the limited dataset variety and forecasting horizons in the original submission. To directly and comprehensively address these concerns, we have conducted a substantially expanded set of experiments in the revised manuscript, with the updated results shown below

| Dataset | O | Autoformer |  | Informer | | Reformer   |  | Transformer         | | **Fighter (κ=3)** | |
|--|--|---|--|-|--|--|--|--|--|--|-|
| | | MSE | MAE | MSE | MAE | MSE | MAE | MSE | MAE | MSE | MAE|
|**Electricity**|96|0.474±0.078|0.485±0.041|0.544±0.006|0.546±0.003|0.374±0.004|0.438±0.004|0.388±0.002|0.442±0.002|**0.339±0.007**|**0.417±0.004**|
| |192|0.581±0.157|0.533±0.074|0.652±0.062|0.616±0.032|0.387±0.010|0.447±0.007|0.384±0.019|**0.441±0.013**|**0.376±0.034**|0.448±0.027|
| |336|0.463±0.078|0.484±0.038|0.686±0.015|0.632±0.005|0.466±0.013|0.502±0.009|0.441±0.053|0.487±0.042|**0.408±0.015**|**0.477±0.012**|
| |720|0.559±0.194|0.539±0.105|0.966±0.007|0.801±0.004|0.490±0.014|0.519±0.008|0.442±0.045|0.484±0.033|**0.380±0.010**|**0.451±0.007**|
|**Weather**|96|0.392±0.023|0.444±0.013|0.911±0.095|0.709±0.039|1.076±0.129|0.813±0.054|0.564±0.005|0.545±0.004|**0.270±0.008**|**0.344±0.007**|
| |192|0.348±0.023|0.415±0.020|0.898±0.036|0.708±0.017|1.080±0.093|0.806±0.041|0.620±0.013|0.569±0.010|**0.256±0.012**|**0.330±0.009**|
| |336|0.384±0.037|0.430±0.028|0.930±0.020|0.729±0.010|1.038±0.019|0.788±0.002|0.669±0.102|0.585±0.050|**0.296±0.008**|**0.358±0.007**|
| |720|0.628±0.040|0.587±0.027|1.137±0.025|0.831±0.014|0.693±0.137|0.614±0.073|0.957±0.078|0.724±0.035|**0.342±0.004**|**0.383±0.004**|
|**ETTh1**|96|1.117±0.043|0.846±0.010|1.135±0.049|0.799±0.016|1.139±0.016|0.791±0.009|0.919±0.040|0.754±0.022|**0.732±0.076**|**0.651±0.050**|
| |192|1.148±0.085|0.839±0.026|1.108±0.060|0.793±0.028|1.251±0.035|0.840±0.014|0.949±0.016|0.762±0.007|**0.727±0.300**|**0.631±0.164**|
| |336|1.156±0.031|0.867±0.006|1.246±0.033|0.883±0.019|1.383±0.016|0.884±0.004|0.971±0.058|0.778±0.031|**0.912±0.242**|**0.759±0.135**|
| |720|1.108±0.044|0.846±0.019|1.267±0.019|0.873±0.010|1.425±0.009|0.894±0.003|1.049±0.007|0.810±0.009|**0.897±0.163**|**0.763±0.096**|
|**ETTm1**|96|0.903±0.004|0.736±0.007|0.916±0.044|0.718±0.027|1.034±0.015|0.744±0.009|0.572±0.046|0.542±0.024|**0.500±0.032**|**0.499±0.021**|
| |192|0.877±0.078|0.728±0.046|0.955±0.029|0.727±0.014|1.196±0.017|0.807±0.005|0.690±0.046|0.609±0.018|**0.517±0.036**|**0.514±0.034**|
| |336|0.999±0.014|0.788±0.031|1.046±0.068|0.771±0.031|1.391±0.018|0.885±0.006|0.807±0.014|0.675±0.010|**0.603±0.066**|**0.572±0.041**|
| |720|0.998±0.105|0.787±0.051|1.090±0.022|0.795±0.005|1.538±0.001|0.926±0.001|1.002±0.030|0.751±0.003|**0.701±0.026** | **0.627 ± 0.029** |

| Dataset|O|Autoformer|Informer|Reformer|Transformer|**Fighter(κ=3)**|
|--|--|--|--|--|--|-|
|**AGNews**|--|0.876|0.831|0.820|0.864|**0.898** |

Specifically:

1. We added two widely used benchmarks in the time-series forecasting community: ETTh1 and ETTm1, which are considered standard and particularly challenging due to their high noise and complex seasonality.

2. We extended the prediction horizons on all datasets from the original maximum of 192 steps to 720 steps (i.e., from ~8 days to 30 days ahead for hourly data), covering short-, medium-, and long-term forecasting regimes that are standard in the literature.

3. All results are now reported with mean ± standard deviation over three random seeds, eliminating any concern about missing variability or cherry-picking.

These extensions show that Fighter continues to perform competitively across a broader range of datasets and substantially longer horizons. The performance gap relative to baselines tends to remain stable or increase slightly at longer prediction lengths (e.g., O=720 on Electricity and Weather), consistent with our original claims, albeit now supported by a more comprehensive and statistically reported evaluation.

We thank the reviewers again for these suggestions, which have clearly strengthened the empirical validation of the paper.

---

> ### Author Response · Authors · 2025-11-29
> **Part (2/2)**
>
> # Generality of Multi-Hop Graph Aggregation: Improvements on Recent Baselines
>
> We thank the reviewers for suggesting comparisons with the latest baselines such as iTransformer. While our primary goal has never been to claim a new state-of-the-art on every benchmark, we fully agree that evaluating against the strongest recent methods is valuable.
> More importantly, the multi-hop graph aggregation mechanism we propose is designed to be architecture-agnostic and easily transferable. To illustrate its generality and practical utility, we applied it as a lightweight plug-in module to the current leading iTransformer encoder-only model.
>
> As shown in the newly added Table below, this simple integration brings consistent performance gains on both ETTh1 and ETTm1 datasets:
>
> | Horizon |Model|ETTh1(MSE/MAE)|ETTm1(MSE/MAE)|
> |--|--|-|-|
> |96|iTransformer|0.413/0.423|0.367/0.388|
> | | + multi-hop aggregation ($\kappa=3$)  | **0.404 / 0.417** | **0.340 / 0.374** |
> | 720 | iTransformer |0.547/0.516|0.497/0.460|
> |  | + multi-hop aggregation ($\kappa=3$) | **0.522 / 0.505** | **0.488 / 0.454** |
>
> The integration yields consistent improvements on both datasets and both horizons (up to 7.4 % relative MSE reduction) without requiring extra hyperparameter tuning or significant computational overhead. These results confirm that the multi-hop aggregation serves as a simple yet effective enhancement that can benefit recent strong baselines.
>
> # Clarification and Empirical Analysis of Computational and Memory Complexity
>
> We sincerely thank the reviewer for pointing out the insufficient discussion of practical computational cost. We have added a dedicated section (Appendix C Additional Experiment) in the revised manuscript that combines theoretical complexity analysis with concrete measurements.
>
> ## Theoretical complexity (per layer)
>
> The attention matrix $A \in \mathbb{R}^{S \times S}$ is identical to a vanilla Transformer. The additional memory cost of Fighter arises mainly from storing the κ-hop powers and the block-diagonal aggregation, leading to overall memory complexity of $O(\kappa S^2 + \kappa S d)$, i.e., linear in κ.
>
> As correctly derived in the original paper, the standard attention distribution matrix $ A = \mathrm{softmax}(d^{-1/2} QK^\top) \in \mathbb{R}^{S \times S} $ is computed once in $O(S^2 d)$, exactly the same as a vanilla Transformer. The multi-hop extension constructs
> $ A^{[\kappa]} = [I, A, A^2, \dots, A^{\kappa-1}] \in \mathbb{R}^{S \times \kappa S}. $ Naïve power computation would cost $O(\kappa S^3)$, but we adopt iterative multiplication $A^k = A \cdot A^{k-1}$, which reduces the cost of building all powers to $O(\kappa S^3)$. Subsequent block-diagonal aggregation and linear projection are $O(S \kappa d^2)$. Thus the total per-layer complexity of Fighter is $ O(S^2 d + \kappa S^3 + S \kappa d^2). $  In practice (long-sequence forecasting), $d$ is small (64–512) while $S$ ranges from hundreds to several thousand, making the $\kappa S^3$ term dominant when $\kappa$ becomes large.
>
> ## Empirical resource consumption (Weather dataset, pred len = 96)
>
> Table 1: Peak GPU memory comparison
>
> | Model | Autoformer | Informer | Reformer | Vanilla Transformer | Fighter (κ=3) |
> |--|--|-|--|--|-|
> | Peak GPU Memory (MB) |108.0|120.2|105.6|109.1|157.3 |
>
> Table 2: Memory scaling of Fighter with hop number κ
>
> | κ | Peak GPU Memory (MB) | Relative to κ=1 |
> |---|--|--|
> |1|94.5|1.00×|
> |2|125.4|1.33×|
> |3|157.3|1.66×|
> |4|188.8|2.00×|
> |5|233.2|2.47×|
> |6|278.7|2.95×|
> |7|337.1|3.57×|
>
> Fighter ($\kappa=3$) implements the multi-hop graph aggregation, which enables long-range information to propagate naturally by higher-order compositions of the attention distribution matrix. As expected, it consumes relatively more GPU memory than the baselines but obtains superior performance. Furthermore, when $\kappa=1$, Fighter exactly recovers a simplified version of the vanilla Transformer in which (i) the attention distribution matrix effectively collapses to the identity matrix, and (ii) the redundant second linear projection of the feed-forward network is removed (as justified by our GCN equivalence). Consequently, Fighter ($\kappa=1$) consumes noticeably less GPU memory than the vanilla Transformer. This confirms that the extra memory cost of Fighter is incurred only by the multi-hop aggregation branches ($\kappa\geq 1$) and that, even at $\kappa=1$, our streamlined design already brings a small but non-trivial memory saving without any performance degradation. Notably, the overhead is linear in $\kappa$, as presented in Table 2. Although Fighter($\kappa=3$) introduces additional computational branches, resulting in a moderate increase in GPU memory usage, the overall consumption remains lightweight and well within practical limits. This overhead is justified by the substantial performance gains Fighter delivers compared with conventional Transformer-based models.

---

### Author Response · Authors · 2025-11-30
**Author Final Remarks**

Dear Area Chair,

Thank you for handling our submission. We greatly appreciate the reviewers’ careful, constructive, and highly detailed feedback. We have carefully addressed every point raised by the reviewers through extensive new experiments, clearer explanations, additional ablations, complexity analyses, and an expanded related work section, all of which are now fully reflected in the revised manuscript and in the detailed rebuttal.

## Summary of Rebuttal

Since notification, we have already:

- Fixed the train/val/test split inconsistency and re-run every single experiment with identical splits and three random seeds.

- Added the two most standard long-term forecasting benchmarks (ETTh1 & ETTm1).

- Extended all horizons up to 720 steps.

- Reported full mean ± std over three seeds.

- Added a thorough complexity analysis (theoretical + measured GPU memory & scaling with κ).

- Added a new ablation showing that our multi-hop aggregation is a simple plug-in that consistently improves the  current leading iTransformer encoder-only model (up to 7.4 % relative MSE reduction).

- Added gradient-norm stability curves, deeper layer scaling ablations, and extended related-work discussion that clearly differentiates us from spatial-graph forecasting models and from earlier Transformer-GCN analogies.

## Summary of Contributions and Significance of Fighter

We are the first work to:

1. Establish a rigorous forward + backward equivalence between Transformer encoders and (flexible) GCNs, including exact gradient derivations for the dynamic structure-learning parameters W_Q and W_K.

2. Use this equivalence to expose and safely remove a genuine redundancy in the standard Transformer blocks (the second FFN projection), yielding a cleaner, more interpretable architecture.

3. Introduce explicit multi-hop aggregation on the attention-derived dynamic adjacency matrix, giving controllable long-range temporal propagation while remaining orthogonal to multi-head attention.

4. Show that the resulting Fighter architecture is both more performant and more interpretable than vanilla Transformers and competitive with (or superior to) heavily engineered forecasting models, while the multi-hop mechanism transfers as a lightweight improvement to current leading models such as iTransformer.

Although we were unable to engage in further discussion with the reviewers due to the current ICLR policy, we are fully confident that the current version resolves all major concerns raised (limited datasets/horizons, missing variability, complexity discussion, comparison to recent baselines, and depth of the theoretical contribution). We believe these combined theoretical and empirical advances make Fighter a meaningful contribution worthy of inclusion at ICLR.

Thank you again for the constructive and high-quality reviews.

Respectfully,

The Authors

---

### Meta-Review · Area_Chair_bdqk · 2026-01-07

**Summary:**

Reviewers have found a few major issues. The most important one is the existence of prior work on reporting similar phenomena as done in this article. However, the most important part of the reviews comes from the reviewer RRhp, who questions the results.

In general, the responses in this rebuttal are not satisfactory and do a poor job. Instead of responding to the reviewers' concerns, the authors have prepared general responses that fail to adequately address them. For example, RRhp is asking for source code, and the response does not mention it. The reviewer is also asking about performance issues, and the authors indirectly acknowledge only some mistakes: "We discovered that the train/validation/test splits were not identical across different methods and ablations in the original experiments".

All the results, hence, are subject to change and require a complete reread of the reviewers, which will not happen at this late stage of changes.

**Reviewer Concerns:**

HWZj
- Additional datasets and ablation studies.
-Stability and convergence analysis.

rpL3
- The article does not position the work with respect to existing transformer models.
- Computational complexity analysis is missing.

SazB
- Asks for the difference in attention matrices. The authors do not report the difference in the response ("we computed the Frobenius norm of the difference", but where is this?)
- A table for comparing the number of parameters. The authors do not give the table, but discuss that it is not applicable.

RRhp
- Novelty is limited because the phenomena had been reported earlier in a blog post and an article.
- The reviewer mentions that training hyperparameters are not reported, and the code is not shared.
- Finds major doubtful performance issues. The authors do not even directly answer these questions. This issue alone is a reason for rejection.

**Reviewer Scores:**

Hwzj rated 4, rpL3 rated 4, but they are cautious about the article's novelty and especially the experimental discussion. SazB rated 6, and with directed responses, may have changed their rating; however, the responses are not satisfactory.
RRhp rated 2, and would definitely not increase their score.

---

### Decision · Program_Chairs · 2026-01-26

Reject